# Boosting Sample Efficiency and Generalization in Multi-agent Reinforcement Learning via Equivariance

**Joshua McClellan**
JHU APL *
University of Maryland
joshmccl@umd.edu

**Naveed Haghani**
JHU APL
naveed.haghani@jhuapl.edu

**John Winder**
JHU APL
john.winder@jhuapl.edu

**Furong Huang**
University of Maryland
furongh@umd.edu

**Pratap Tokekar**
University of Maryland
tokekar@umd.edu

## Abstract

Multi-Agent Reinforcement Learning (MARL) struggles with sample inefficiency and poor generalization [1]. These challenges are partially due to a lack of structure or inductive bias in the neural networks typically used in learning the policy. One such form of structure that is commonly observed in multi-agent scenarios is symmetry. The field of Geometric Deep Learning has developed Equivariant Graph Neural Networks (EGNN) that are equivariant (or symmetric) to rotations, translations, and reflections of nodes. Incorporating equivariance has been shown to improve learning efficiency and decrease error [2]. In this paper, we demonstrate that EGNNs improve the sample efficiency and generalization in MARL. However, we also show that a naive application of EGNNs to MARL results in poor early exploration due to a bias in the EGNN structure. To mitigate this bias, we present *Exploration-enhanced Equivariant Graph Neural Networks* or E2GN2. We compare E2GN2 to other common function approximators using common MARL benchmarks MPE and SMACv2. E2GN2 demonstrates a significant improvement in sample efficiency, greater final reward convergence, and a 2x-5x gain in over standard GNNs in our generalization tests. These results pave the way for more reliable and effective solutions in complex multi-agent systems.

## 1 Introduction

Multi-Agent Reinforcement Learning (MARL) has found success in various applications such as robotics [3, 4, 5], complex strategy games [6, 7, 8] and power grid management [9, 10]. However, MARL algorithms can be slow to train, difficult to tune, and have poor generalization guarantees [1, 11]. This is partially because typical implementations of MARL techniques use neural networks such as Multi-Layer Perceptrons (MLP) that do not take the underlying

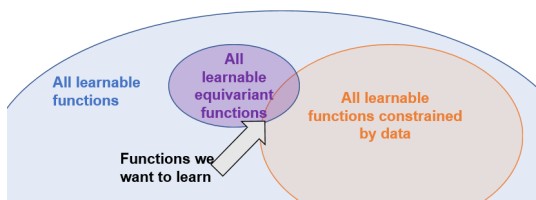

Figure 1: An example of how using an equivariant function approximator shrinks the total search space.

structure into account. The models learn simple input/output relationships with no constraints or priors on the policies learned. These generic architectures lack a strong inductive bias making them inefficient in terms of the training samples required.

---

*The Johns Hopkins University Applied Physics Lab

38th Conference on Neural Information Processing Systems (NeurIPS 2024).

Symmetries are commonplace in the world. As humans, we exploit symmetries in our daily lives to improve reasoning and learning. It is a basic concept learned by children. Humans do not need to relearn how to eat an apple simply because it has been moved from the right to the left. In soccer, if you have learned to pass to someone on the right, it is easier to learn to pass to someone on the left. Our objective is to develop agents that are guaranteed to adhere to these basic principles, without needing to learn every single scenario from scratch.

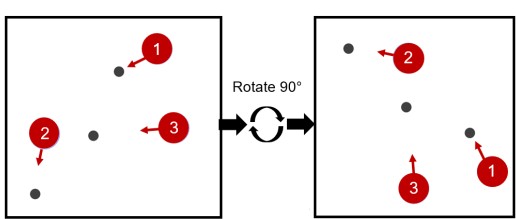

Figure 2: An example of rotational equivariance/symmetry in MPE simple spread environment. Note as the agent (in red) positions are rotated, the optimal actions (arrows) are also rotated.

Symmetries are particularly common in MARL. These occur in the form of *equivariance* and *invariance*. Given a transformation matrix $\boldsymbol{T}$, if $f(\boldsymbol{T}x) = f(x)$ that function is said to be invariant. Similarly, $f(.)$ is equivariant if $f(\boldsymbol{T}x) = \boldsymbol{T}f(x)$ [12]. Rotational and reflection symmetries (the $O(n)$ symmetry group) are particularly common in Reinforcement Learning (RL) scenarios. For example, consider the Multi-agent Particle Environment (MPE) [13] benchmark for MARL which has agents with simple dynamics and tasks. We note that this scenario adheres to rotational equivariance. As shown in Figure 2 rotating the agent's positions results in the optimal actions also being rotated. A policy that is rotationally equivariant effectively shrinks the state-action space for the problem, potentially making the problem easier to learn (Figure 1).

One way to guarantee equivariance to rotations and reflections in MARL is to use an Equivariant Graph Neural Network [2]. Due to the more complex and equivariant structure of the EGNN, directly adapting EGNNs to MARL is not straightforward. Specifically, consider action spaces typically represented as discrete choices (e.g., up, down, left, shoot). Typically the policy over such an action space is represented using logits (to specify the probability of choosing each action). EGNN outputs a continuous equivariant output. Mapping this output to logits is non-trivial. Therefore, we ask the question *What is the correct representation of a complex action space when using Equivariant GNNs?*

Another specific issue we observed in EGNNs is an exploration bias in the early stages of learning due to the specific nature of the equivariant computation. This bias can lead to poor initial exploration, decreasing the probability of the agent finding the optimal policy, and potentially resulting in sub-optimal convergence. Therefore we ask the question *How can we design a GNN equivariant to rotations, but without early exploration bias?*

The main contribution of this paper is Exploration-enhanced Equivariant Graph Neural Networks (E2GN2). This addresses the two research questions. Specifically, we show how to apply Equivariant GNNs to complex MARL tasks, to mixed discrete/continuous action spaces, and to mitigate the early exploration bias. Our contributions can be summarized as:

(1) Our approach is the first to successfully demonstrate Equivariant GNNs for MARL on standard MARL benchmarks with complex action spaces.

(2) We propose E2GN2 which has no bias in early exploration for MARL and is equivariant to rotations and reflections.

(3) We evaluate E2GN2 on common MARL benchmarks: MPE [13] and Starcraft Multi-agent Challenge v2 [14] using PPO. It learns quicker, outperforming standard GNNs and MLPs by up to 2x-5x [2] in sample efficiency on terran and protoss challenges. It is worth noting that E2GN2 is an improvement on the function approximation for MARL and thus is compatible with most MARL actor-critic methods.

(4) We showcase E2GN2's ability to generalize to scenarios it wasn't trained on, due to the equivariance guarantees. This results in 5x performance over GNNs

---

[2]For terran, E2GN2 reaches 0.4 win rate after about $1.25 \times 10^6$ timesteps, GNN reaches ~0.4 after $5 \times 10^6$ timesteps and MLP maxes out at ~0.3 win rate after $10 \times 10^6$ timesteps. For Protoss, E2GN2 reaches 0.4 win rate after $2.5 \times 10^6$ timesteps, and GNN and MLP do not reach 0.4 within $10 \times 10^6$ timesteps. See figure 6.

## 2 Related Works

A key theoretical foundation for our paper is [15], which formulated the structure for equivariant MDPs. One important takeaway is that if a reward is equivariant to a transformation/symmetry, we want the policy and dynamics to be equivariant to that symmetry. However, this work was limited to very simple dynamics with discrete actions such as cart-pole. [16] followed up with equivariance in multi-agent, but was again limited to simple problems. Their method is specifically formulated and tested on small discrete grid world problems with simple dynamics and discrete up, down,left actions. For example, the trafic control problem has an input of a 7x7 grid. Extending their work to continuous environments with large state spaces, large mixed discrete/continuous action spaces is not straightforward without significant modifications.

Contemporary to our research, [17] demonstrated E(3) equivariant networks on simple cooperative navigation problems. However, their results on more complex tasks, such as Starcraft, did not excel over the baseline. Additionally, they used SEGNN [18] which can result in very slow training times, making tuning difficult and cumbersome. Others took the approach [19] of attempting to learn symmetries via an added loss term. However, since this approach needed to learn the symmetries in parallel with learning it did not have the same guarantees as Equivariant GNNs, and did not result in significant performance gains. Another work [20] demonstrated rotation equivariance for complex robotic manipulation tasks. This work, while promising, was for single-agent RL and used image observations and many problems don't have access to image observations.

AI research in chemistry has taken a particular interest in adding symmetry constraints to GNNs. Works such as EGNN, [2] SEGNN, [18] E3NN, [21] and Equivariant transformers have demonstrated various approaches to encoding symmetries in GNNs. In this paper, we chose to focus on EGNN due to its simplicity, high performance, and quick inference time. Other works took a different approach, such as [22], which proposes Equivariant MLPs, solving a constrained optimization problem to encode a variety of symmetries. Unfortunately, in our experience, the inference time was slow, and we preferred a network with a graph structure such as EGNN.

## 3 Background

### 3.1 MARL

Multi-agent reinforcement learning (MARL) considers problems where multiple learning agents interact in a shared environment. The goal is for the agents to learn policies that maximize long-term reward through these interactions. Typically, MARL problems are formalized as Markov games [23]. A Markov game for $N$ agents is defined by a set of states $\mathcal{S}$, a set of actions $\mathcal{A}_1, ..., \mathcal{A}_N$ for each agent, transition probabilities $P : \mathcal{S} \times \mathcal{A}_1 \times ... \times \mathcal{A}_N \times \mathcal{S} \to [0, 1]$, and reward functions $R_1, ...R_N$ mapping each state and joint action to a scalar reward.

The goal of each agent $i$ is to learn a policy $\pi_i(a_i|s)$ that maximizes its expected return: $J(\pi_i) = \mathbb{E}\pi_1, ..., \pi_N \left[ \sum_{t=0}^{T} \gamma^t R_i(s_t, a_t^1, ..., a_t^N) \right]$

Where $T$ is the time horizon, $\gamma \in (0, 1]$ is a discount factor, and $a_t^j \sim \pi_j(\cdot|s_t)$. The presence of multiple learning agents makes this more complex than single-agent RL due to issues such as non-stationarity and multi-agent credit assignment.

### 3.2 Equivariance

Crucial to equivariance is group theory. A group is an abstract algebraic object describing a symmetry. For example, $O(3)$ describes the set of continuous rotation symmetries. A group action is an element of that particular group. To describe how groups o perate on data we use representations of group actions. A representation can be described as a mapping from a group element to a matrix, $\rho : G \to GL(m)$ where $\rho(g) \in \mathbb{R}^{m \times m}$ Or instead we can more simply use: $L_g : X \to X$ where $g \in G$ where $L_g$ is the matrix representation of the group element $g \in G$ [24].

A function is equivariant to a particular group or symmetry if transforming the input is equivalent to transforming the function output. More formally, $T_g f(x) = f(L_g x)$ for $g \in G$, $L_g : X \to X$ and $T_g : Y \to Y$. Related to equivariance is the key concept of invariance, that is a function does not change with a transformation to the input: $f(x) = f(L_g x)$. [24]

Previous work [16] has shown that if a Markov game has symmetries in the dynamics and the reward function then the resulting optimal policy will be equivariant and the value function will be invariant. That is, $V(L_g\boldsymbol{s}) = V(\boldsymbol{s})$ and $\pi(L_g\boldsymbol{s}) = K_g\pi(\boldsymbol{s})$ where $L_g$ and $K_g$, with $g \in G$ are transformations to the state and action respectively.

## 3.3 Equivariant Graph Neural Network

Equivariant Graph Neural Networks [2] (EGNN) are an extension of a standard Graph Neural Network. EGNNs are equivariant to the $E(n)$ group, that is, rotations, translations, and reflections in Euclidean space. EGNNs have two vectors of embeddings for each graph node $i$: the feature embeddings denoted by $\boldsymbol{h}_i^l$, and coordinate embeddings denoted by $\boldsymbol{u}_i^l$, where $l$ denotes the neural network layer. The equations describing the forward pass for a single layer are below:

$$\boldsymbol{m}_{ij} = \phi_e\left(\boldsymbol{h}_i^l, \boldsymbol{h}_j^l, \|\left(\boldsymbol{u}_i^l - \boldsymbol{u}_j^l\right)\|^2\right) \tag{1}$$

$$\boldsymbol{u}_i^{l+1} = \boldsymbol{u}_i^l + C\sum_{j\neq i}\left(\boldsymbol{u}_i^l - \boldsymbol{u}_j^l\right)\phi_u\left(\boldsymbol{m}_{ij}\right) \tag{2}$$

$$\boldsymbol{m}_i = \sum_{j\neq i}\boldsymbol{m}_{ij}, \quad \boldsymbol{h}_i^{l+1} = \phi_h\left(\boldsymbol{h}_i^l, \boldsymbol{m}_i\right) \tag{3}$$

Here $\phi$ represents a multi-layer perceptron, where $\phi_e : \mathbb{R}^n \mapsto \mathbb{R}^m$, $\phi_u : \mathbb{R}^m \mapsto \mathbb{R}$, and $\phi_h : \mathbb{R}^{m+p} \mapsto \mathbb{R}^p$. The key difference between EGNN and GNN is the addition of coordinate embeddings $\boldsymbol{u}_i$ and equation 2, which serves to update the coordinate embeddings in a manner that is equivariant to transformations from $E(n)$. Note that $\boldsymbol{u}_i$ will be equivariant and $\boldsymbol{h}_i$ will be invariant to these transformations [2].

As noted in Section 1, application of EGNN to MARL is not straightforward. In the following section, we discuss these issues in more depth and present our solution towards addressing them.

# 4 Methods

In this section, we address both theoretically and empirically how the output of EGNN is biased by the input, leading to suboptimal exploration in RL. To mitigate this issue, we introduce Exploration-enhanced Equivariant Graph Neural Networks (E2GN2) as a method that ameliorates this bias, leading to improved exploration.

## 4.1 Biased Exploration in EGNN Policies

An important component of reinforcement learning is exploration. Practitioners often use a policy parameterized by a Gaussian distribution, where the mean is determined by the policy network output, and the standard deviation is a separately learned parameter. Best practices are that the actions initially have a zero mean distribution to get a good sampling of potential state-action trajectories, i.e., $\pi(\boldsymbol{a}_i|\boldsymbol{s}) \sim N(\boldsymbol{0}, \sigma)$. Below we show that an EGNN will initially have a non-zero mean distribution, which can cause problems in early training.

**Theorem 1** *Given a layer $l$ of an EGNN with randomly initialized weights, with the equivariant component input vector $\boldsymbol{u}_i^l \in \mathbb{R}^n$, equivariant output vector $\boldsymbol{u}_i^{l+1} \in \mathbb{R}^n$ and the multi-layer perceptron $\phi_u : \mathbb{R}^m \mapsto \mathbb{R}$, where the equivariant component is updated as: $\boldsymbol{u}_i^{l+1} = \boldsymbol{u}_i^l + C\sum_{j\neq i}\left(\boldsymbol{u}_i^l - \boldsymbol{u}_j^l\right)\phi_u\left(\boldsymbol{m}_{ij}\right)$. Then the expected value of the output vector is approximately the expected value of the input vector:*

$$\mathbb{E}\left[\boldsymbol{u}_i^{l+1}\right] \approx \mathbb{E}\left[\boldsymbol{u}_i^l\right]$$

*Furthermore, given a full EGNN with L layers then the expected value of the network output is approximately the expected value of the network input*

$$\mathbb{E}\left[\boldsymbol{u}_i^L\right] \approx \mathbb{E}\left[\boldsymbol{u}_i^0\right]$$

(See appendix A for proof.)

**Corollary 1.1** *Given a policy for agent $i$ represented by an EGNN and parameterized with a Gaussian distribution, and the equivariant component of the state $\boldsymbol{s}_i^{eq}$ the policy will have the following distribution:*

$$\pi_i(\boldsymbol{a}_i|\boldsymbol{s}) \sim N(\boldsymbol{s}_i^{eq}, \sigma)$$

(See Appendix A for proof.)

In many cases $\boldsymbol{s}_i^{eq}$ is the agent's position. Corollary 1.1 indicates that an agent's action distribution will be skewed towards its own position. If the magnitude of the state representation $\boldsymbol{s}_i^{eq}$ is significantly larger than $\sigma$, the agent may output actions approximately equal to its position. We refer to this phenomenon, where agent actions are biased towards replicating the input state as the action, as the *early exploration bias*. Such bias is not conducive to effective exploration, potentially limiting the agent's ability to discover and converge to optimal solutions.

**Examples of early exploration bias** Suppose there is an agent placed at the position $\boldsymbol{u}_i = [2, 0]$, and its objective is to reach the origin (i.e. $\boldsymbol{0}$). The agent needs an action of less than 0 to move to the left. Since its action using the EGNN is close to Gaussian distributed with a mean of 2, then $P(X < 0) \approx 0.02$. If the agent moves 0.25 with each action, then the probability of the agent reaching the center objective in eight steps is 2.9e-8. On the other hand, the agent has a high likelihood of moving toward infinity as each move away increases the likelihood of moving towards infinity. This example shows that the bias leads to a low probability of finding the solution, which is especially harmful in sparse reward environments.

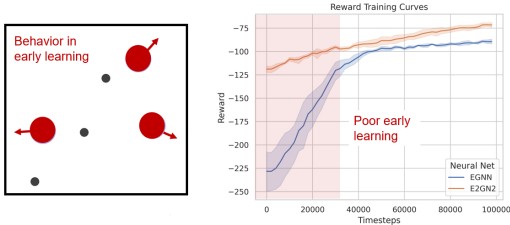

Figure 3: An example of biased learning in MPE simple spread environment. **Left**: We observed the behavior of the EGNN agents in this early training phase. Each agent moved away from the origin due to the EGNN bias. **Right**: Note the very low reward in early training steps due to the biased policies moving away from the goals.

To confirm our hypothesis on EGNN early exploration bias, we conducted a simple experiment. We trained a Proximal Policy Optimization (PPO) [3] agent on the MPE simple-spread problem. This problem consists of three agents that seek to navigate to three goals cooperatively; the agents are rewarded based on the sum of the minimum distance from each target to an agent. We give further details of this set up and training procedure in the experiments section. Figure 3 shows the results. The EGNN policy biased the agents to move in the direction of their current position, causing them to move away from the origin and receive low rewards. Due to the dense rewards and simplicity of the problem, the EGNN agent was able to overcome this initial bias and still solve the problem. However, in more complex problems this early bias could cause more problems. Even if an agent finds an optimal trajectory, it will be a relatively small proportion of the sampled trajectories and may result in sub-optimal convergence.

## 4.2 Exploration-Enhanced EGNNs

As discussed previously, EGNN's severe early exploration bias can decrease learning performance. In this section, we propose our solution to this problem in the form of Exploration-enhanced Equivariant Graph Neural Networks (E2GN2). To create E2GN2 we make the following modification to Equation 2 of the equivariant component of EGNN:

$$\boldsymbol{u}_i^{l+1} = \boldsymbol{u}_i^l \phi_{u_2}(\boldsymbol{m}_i) + C \sum_{j \neq i} \left( \boldsymbol{u}_i^l - \boldsymbol{u}_j^l \right) \phi_u(\boldsymbol{m}_{ij})) \tag{4}$$

where $\phi_{u_2}(\boldsymbol{m}_i) : \mathbb{R}^m \to \mathbb{R}$ is an MLP parameterized by $u_2$. The remaining layer update equations of the EGNN remain the same.

**Theorem 2** *Given a $L$ layer E2GN2 with randomly initialized weights, where the equivariant component is updated as: $\boldsymbol{u}_i^{l+1} = \boldsymbol{u}_i^l \phi_{u2}(\boldsymbol{m}_i) + C \sum_{j \neq i} \left( \boldsymbol{u}_i^l - \boldsymbol{u}_j^l \right) \phi_u(\boldsymbol{m}_{ij}))$. Then the expected value of the output vector is approximately the expected value of the input vector:*

$$\mathbb{E}\left[\boldsymbol{u}_i^{l+1}\right] \approx \boldsymbol{0}$$

**Corollary 2.1** *Given a policy for agent $i$ represented by an E2GN2 and parameterized with a Gaussian distribution the policy will have the following distribution:*

$$\pi_i(\boldsymbol{a}_i|\boldsymbol{s}) \sim N(\boldsymbol{0}, \sigma)$$

Thus we see that E2GN2 should have unbiased early exploration for the equivariant component of the actions. The primary difference between EGNN and E2GN2 is the addition of $\phi_{u_2}$, which serves to offset the bias from the previous layer (or input) $u_i^l$ To validate that this did indeed solve the exploration bias we tested E2GN2 on the same MPE simple spread environment. We observe in Figure 3 that E2GN2 did indeed have unbiased behavior when it came to early exploration, as the agents had smooth random actions, and the reward did not drastically decrease.

**Analysis of E2GN2 Equivariance**    Here we verify that E2GN2 still retains EGNN's guarantee of equivariance to rotations and reflections.

**Theorem 3** *E2GN2 is equivariant to transformations from the $O(n)$ group. In other words, it is equivariant to rotations and reflections.*

(See appendix A for proof.)

Retaining our symmetries to rotations and reflections is important, since it should increase our sample efficiency and generalization. Note that E2GN2 is no longer equivariant to translations. However, in MARL translation equivariance is not necessarily a benefit. For example, consider the MPE problem with a translational equivariant policy. If we shift an agent to be 10 units right, this will add 10 to the action as well, causing it to move to the right! Essentially, this is adding an undesirable bias to the policy output: $\pi(s + b) = \pi(s) + b$ However, we can expect $O(n)$ policy equivariance to improve our sample efficiency in MARL, and it is key that E2GN2 retains this guarantee.

### 4.3   Adapting Architectures for Complex Action Spaces

Applying EGNN/E2GN2 to sophisticated MARL problems requires careful consideration. Many MARL environments have discrete action spaces or mixed continuous-discrete action spaces. Some components of these action spaces may be invariant and others may be equivariant. Further complicating the problem, MARL requires the neural networks to output parameters for a probability distribution (which is sampled for use in exploration). For continuous actions, agents typically use a gaussian and for discrete actions agents generally use logits. Unfortunately, it is not straightforward to map the distinct invariant and equivariant coordinate embeddings onto a single distribution in a manner that preserves equivariance.

A core benefit of the GNN structure in MARL is the scalability afforded by permutation equivariance: the ability to handle a variable number of agents without retraining. For example, [25] demonstrates using a GNN to train $N$ agents, then to control $N + 1$ agents without retraining. However, this example did not operate with discrete or mixed action spaces or equivariant structures as we do here. Improperly mapping GNN outputs to logits risks losing this scalability.

To address these issues, we propose leveraging the GNN's graph structure to output different components of the action space from different nodes in an equivariant manner:

- **Discrete Actions & Invariant Components:** the invariant feature embeddings $h_i$ of each agent's node are used to output discrete logits/actions and other invariant components. For action spaces with both equivariant and invariant components, this can be used as the 'action type selector' to select which type of action to apply at that type step (ie move action or targeting action).

- **Continuous Spatial Actions:** the equivariant coordinate embeddings $u_i$ of each agent's node are used for continuous spatial actions such as movement.

- **Targeting Specific Entities:** for multi-agent environments with entities beyond the learning agents (e.g., enemies), logits for discrete actions pertaining to each entity (e.g., which enemy to target) are output from that entity's corresponding node. This enables the discrete action space to scale with the number of agents.

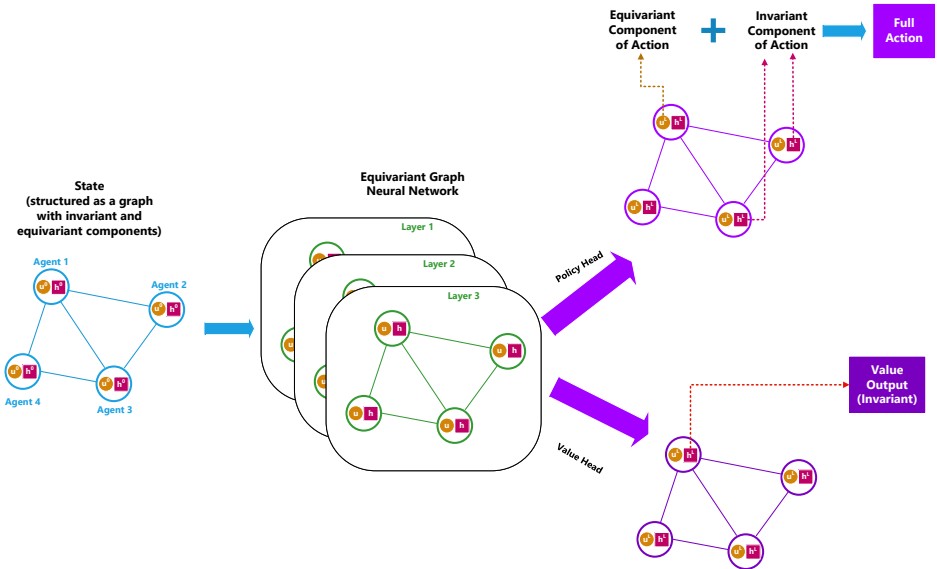

Figure 4: An example of using an Equivariant Graph Neural Network in MARL. Note that the state must be structured as a graph with each node having an equivariant $u_i$ and invariant $h_i$ component. As discussed in 4.3, the output of the policy uses $u_i$ for equivariant (typically spatial) actions, and the $h_i$ for invariant components of the actions

Each of these components is parameterized as a distribution (ie logits or gaussian), resulting in one to three seperate distributions (depending on the environment and which components are used). After the exploration phase samples from each of these distributions, the final action can be constructed from each of these components via concatenation or the action selection component.

This formulation enables us to retain the permutation equivariance and scalability of a GNN. An alternative to our approach is to add an MLP at the end of the GNN to convert the GNN outputs to the mixed action spaces. This will lose the scalability/locality of the neural network. For example, if you add two more agents how do you modify the final MLP to expand the action space? In our formulation, whenever an new entity $i$ is added to the environment, we can simply add the node $N+1$ to the graph. The action space of the GNN will now be supplemented with $\boldsymbol{u}_{N+1}$ and $\boldsymbol{h}_{N+1}$, allowing us to expand the action space without retraining.

By structuring the GNN output in this manner, we can handle discrete or mixed discrete-continuous action spaces while retaining the equivariance of EGNN/E2GN2 and the flexibility of GNNs to a variable number of agents and entities. This approach allows MARL agents based on equivariant GNNs to be applied to challenging environments with minimal restrictions on the action space complexity. Indeed this approach was key to enabling the success of EGNN/E2GN2 on SMACv2.

## 5 Experiments

We seek to answer the following questions with our experiments: *(1) Do rotationally equivariant policy networks and rotationally invariant value networks improve training sample efficiency? (2) Does E2GN2 improve learning and performance over EGNN? (3) Does Equivariance indeed demonstrate improved generalization performance?*

To address these questions, we use common MARL benchmarks: the multi-agent particle-world environment (MPE) [13] and Starcraft Multi-agent Challenge version 2 (SMACv2) [14]. Our experiments show that equivariance does indeed lead to improved sample efficiency, with E2GN2 in particular performing especially well. We also demonstrate that generalization is guaranteed across rotational transformations.

We want to focus our experiments on the neural networks' effects on MARL performance. To isolate the impact of the network architecture, we avoid using specialized tips and tricks sometimes employed in MARL [26]. This allows us to demonstrate that our proposed networks can improve performance

without relying on these additional techniques. Thus we use a common standardized open source MARL training library RLlib [27]. We use the default multi-agent PPO algorithm, which does not use a centralized critic. We followed the basic hyperparameter tuning guidelines set forth in [26]. That is, we use a large training batch and mini-batch size, low numbers of SGD iterations, and a small clip rate. Further hyperparameter details are found in appendix B.

We compare our results with common neural networks used for MARL benchmarks: multi-layer perceptrons (MLP), and GNNs (we use a GNN structure similar to equations 1, 3, see appendix B for details ). We also compare with the approach from [17] which we refer to as E3AC in our plots. Recall that E3AC also uses neural networks that guarantee equivariance: SEGNN. We integrated E3AC into RLLIB to ensure the RL training procedure remained consistent across our comparisons. Note that the majority of MARL papers on SMAC/SMACv2 use MLPs as the base network. We also compare with GNNs to show the improvement is not primarily due to the graph structure. For the main paper results, we assume full observability since we have not explicitly tackled the partial observability problem yet. In future work, we will seek to remedy this. However, to be thorough we performed experiments with partial observability, resulting in surprising success using E2GN2 (see appendix C).

## 5.1   Training Environments

From the MPE environment ([13]), we use two environments to benchmark our performance: co-operative navigation also known as spread, and predator-prey, also known as tag. The MPE tag environment consists of three RL-controlled agents seeking to catch a third evader agent. For easier comparison, we use a simple heuristic algorithm to control the evader agent. The evader simply moves in the opposite direction of the pursuers. The other MPE environment is the cooperative navigation or simple spread. In this environment, each agent seeks to minimize the distance between all obstacles. To better test equivariance, for the MPE environments we use continuous actions instead of the default discrete actions.

Next we test on SMACv2, a much more difficult environment than MPE. In SMACv2, the units are heterogenous with different capabilities (with different attack ranges, total health, and sometimes action spaces). The unit types are randomized at the beginning of the scenario. The actions include more components than simply movement (such as in MPE), agents can move and attack. The goals are more complex as well. Instead of simply navigating cooperatively as in MPE, the agents must learn attack formations and strategies. Sometimes it may be optimal to sacrifice health or allies in the purpose of the greater strategic objective. SMACv2 has three basic scenarios defined by the unit types: terran, protoss, and zerg. We use 5 agents for each team, and the initial position configuration termed "surrounded" (see fig 7)

The SMACv2 action space poses an interesting problem for GNN structures. A key advantage of GNNs is the permutation equivariance, which leads to scalability without retraining. The default SMACv2 agents will output simply a discrete action determining movement or target for shooting. For our purposes, we modify the action space to be a mixed action space. This consists of a continuous vector for movement actions, a discrete action determining the attack target, and a boolean determining if the agent should shoot, move or no-op (to be complete, we include plots with a discrete action space in appendix C). As discussed in section 4.3 this will both prove a greater test for our algorithms and allow for the GNN to scale to larger numbers of agents.

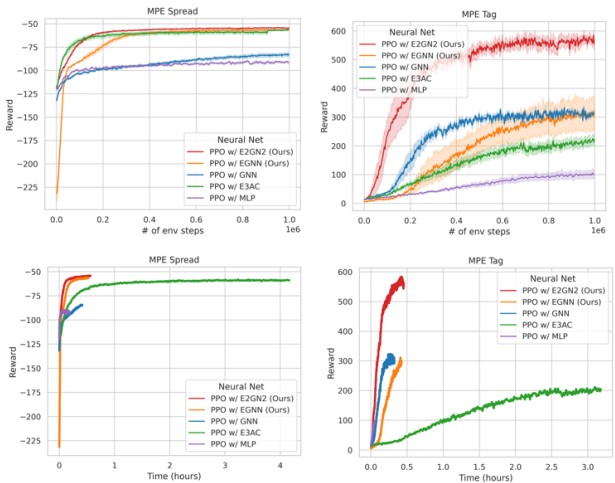

Figure 5: Comparing PPO learning performance on MPE with various Neural Networks. (TOP) reward as a function of environment steps. We show the standard errors computed across 10 seeds. (BOTTOM) reward as a function of wall clock time

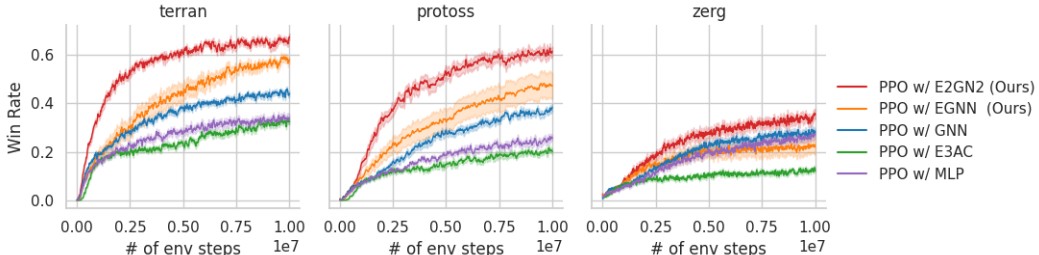

Figure 6: Comparing performance of PPO on SMACv2 with various Neural Networks representing the policy and value function. Each chart represents a different race from the SMACv2 environment. We show the standard errors computed across 10 seeds.

## 5.2 Training Results

We present the results from our training in this section. The results for MPE in Figure 5 are averaged across 10 seeds. As discussed previously, EGNN has poor early performance due to the early exploration bias. Despite this poor exploration, EGNN outperforms GNNs and MLPs, demonstrating the power of equivariance in MARL. Similarly, we note that in MPE tag, E2GN2 results in a strong final agent, while EGNN suffers in the initial training phases. We see that E2GN2 is able to greatly outperform EGNN's final reward, partly due to superior early exploration.

In figure 5 we also compare the wall clock time required to train each algorithm. Note that these were all trained on the same hardware and using the same training algorithm (RLLIBs PPO). E3AC remains competitive in sample efficiency on MPE spread, but it requires nearly four hours to gather one million time steps, compared to less than one hour for E2GN2. This is likely due to the SEGNN [18] underlying structure used by E3AC, which is slower due to the be slower for inference time. We also note that E2GN2 does manage to outperform E3AC on MPE Tag.

Next, we review the results from SMACv2 in Figure 6; these results are averaged across 10 seeds. The equivariant networks have a clear improvment in sample efficiency over the MLP and GNN. On the terran environment, EGNN once again learns slower in the initial phases, but due to equivariance, it outperforms MLP/GNN. For protoss, we note that EGNN performs well but has a high variance for its performance. E3AC struggles to perform well on SMACv2, likely since it was unable to resolve the hurdle we addressed in section 4.3. Instead for SMACv2 E3AC used a simple MLP for the policy and SEGNN for the value function [17]. The training speed and performance for E2GN2 is especially impressive since this environment is much more complex than the MPE.

## 5.3 Generalization

Thus far we have demonstrated that equivariance does indeed lead to improved sample efficiency. The next question to answer is whether equivariance leads to improved generalization. We test this by using the 'surrounded' initial configuration from SMACv2 shown in figure 7. By default, the agents are randomly generated in all directions (along specific axes). To test generalization, we only initialize agents on the left side

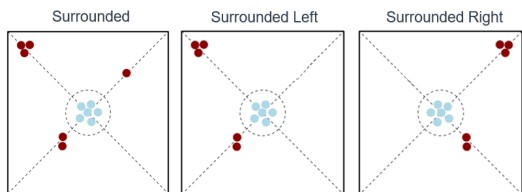

Figure 7: SMACv2 initialization schemes used for testing generalization

of the map (i.e., surrounded left). We then test to see if the agents are able to generalize when they are tested with initial positions starting on the right (called surrounded right), and with the default surrounded configuration (termed Surrounded All or Surrounded). Theoretically, the equivariant network should see equivalent performance between the training and testing initialization, due to the guarantee of rotational equivariance for movement, and invariance for shooting actions.

The results of our tests are shown in Table 1. As expected E2GN2 has stellar generalization performance. The win rate remains the same from the training configuration (Surrounded Left) to Surrounded Right. We see an increase in win rate when testing in Surrounded All. This is likely because this Surrounded All is an easier scenarion; the enemy is divided into more groups, so the agents can defeat the smaller groups one at a time. As we mentioned in the abstract, this table

Table 1: Generalization Win Rate in SMACv2. Note that E2GN2 retains high performance, while GNN and MLP lose performance when generalizing

| Environment | Network | Training Initialization | Testing Initialization | |
| | | Surrounded Left | Surrounded Right | Surrounded All |
|---|---|---|---|---|
| Terran | E2GN2 | $0.57 \pm .01$ | $\mathbf{0.55} \pm .01$ | $\mathbf{0.63} \pm .01$ |
| | GNN | $0.51 \pm .02$ | $0.11 \pm .02$ | $0.32 \pm .02$ |
| | MLP | $0.33 \pm .02$ | $0.12 \pm .02$ | $0.24 \pm .02$ |
| Protoss | E2GN2 | $0.59 \pm .01$ | $\mathbf{0.56} \pm .02$ | $\mathbf{0.57} \pm .02$ |
| | GNN | $0.5 \pm .02$ | $0.14 \pm .01$ | $0.32 \pm .01$ |
| | MLP | $0.42 \pm .02$ | $0.17 \pm .02$ | $0.27 \pm .02$ |
| Zerg | E2GN2 | $0.35 \pm .02$ | $\mathbf{0.32} \pm .02$ | $\mathbf{0.29} \pm .02$ |
| | GNN | $0.4 \pm .02$ | $0.07 \pm .01$ | $0.21 \pm .01$ |
| | MLP | $0.21 \pm .02$ | $0.05 \pm .01$ | $0.12 \pm .01$ |

indicates that for the Surrounded Right generalization test E2GN2 has ~5x the performance of GNN and MLP.

Next, in Table 2 we test E2GN2's ability to scale to various numbers of agents without retraining. We use the agents trained in SMACv2 from Figure 6 for these tests. We do not include the MLP agent since it cannot scale up without retraining. Note that the method described in section 4.3 was essential to enable scaling between agents. For example, if we passed the GNN output to a final MLP to map to the discrete action space this wouldn't be scalable to more agents.

The results for scalability confirm that both E2GN2 and GNN are able to maintain performance with larger numbers of agents. As the number of agents gradually increases the win rate does slowly decrease. E2GN2 does seem to have a slightly steeper loss in performance in the Zerg domain, which was also the most difficult domain for the initial training phase. We believe this is due to the higher complexity of the Zerg scenario, with certain unit types (banelings) that can explode to damage many other agents.

## 6 Conclusion and Future Work

In this paper, we have demonstrated that E2GN2 merits strong consideration for many MARL applications. We addressed several important problems including the early exploration bias and how to apply it to complex action spaces. The sample-efficiency has dramatically improved, and there are now guarantees of generalization built into the network. There is still further work to do, such as handling the partial or incomplete symmetries. Furthermore this approach would need to be improved further to be applied to environments with angular momentum and accelerations, which are not included in these benchmarks. Currently, we expect that the improvement gained from E2GN2 in MARL will depend on the amount of rotational symmetry applicable in the observations. We believe that building on this work can be helpful to deploying MARL agents with a greater degree of trust (due to the guarantees). This could be helpful in many fields such as robotics, medicine, and power systems. In summation, the results of this paper provide a solid foundation upon which to build.

Table 2: Generalization Win Rate: Testing RL agents ability to scale to different numbers of agents (originally trained with 5 agents)

| Environment | Network | Training Setup | Testing Setup | | | |
| | | 5 Agents | 4 Agents | 6 Agents | 7 Agents | 8 Agents |
|---|---|---|---|---|---|---|
| Terran | E2GN2 | $0.69 \pm .02$ | $0.65 \pm .02$ | $0.63 \pm .02$ | $0.62 \pm .02$ | $0.54 \pm .04$ |
| | GNN | $0.45 \pm .01$ | $0.44 \pm .01$ | $0.42 \pm .02$ | $0.39 \pm .01$ | $0.32 \pm .02$ |
| Protoss | E2GN2 | $0.62 \pm .03$ | $0.61 \pm .02$ | $0.59 \pm .03$ | $0.47 \pm .04$ | $0.37 \pm .03$ |
| | GNN | $0.39 \pm .02$ | $0.37 \pm .01$ | $0.36 \pm .03$ | $0.30 \pm .02$ | $0.21 \pm .02$ |
| Zerg | E2GN2 | $0.36 \pm .03$ | $0.32 \pm .03$ | $0.31 \pm .01$ | $0.23 \pm .01$ | $0.18 \pm .03$ |
| | GNN | $0.28 \pm .04$ | $0.28 \pm .02$ | $0.25 \pm .03$ | $0.2 \pm .02$ | $0.15 \pm .02$ |

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

# Appendix and Supplemental Material

## A    Proofs

### A.1    EGNN Bias Proof

Here we prove our assertion that the EGNN layers have biased outputs, and thus poor exploration.

We start by showing that $\mathbb{E}\left[\phi_u(m_{ij})\right] \approx 0$. We do this by computing the expected value of a single neural network layer.

An individual neural network layer is defined as $\boldsymbol{y} = W\boldsymbol{x} + \boldsymbol{b}$ where $\boldsymbol{b}$ is a N-by-1 vector, $\boldsymbol{W}$ is a N-by-M matrix and $\boldsymbol{x}$ is a M-by-1 vector. Note that similar to [28] we initialize the elements in $\boldsymbol{W}$ to be mutually independent and share the same distribution. $\boldsymbol{x}$ is initialized similarly. and $\boldsymbol{x}$ and $\boldsymbol{W}$ are independent. Thus for each element (subscript $l$ ) of the equation, we have:

$$\mathbb{E}\left[y_l\right] = \sum \mathbb{E}\left[w_l x_l\right] + \mathbb{E}\left[b_l\right] = n_l \mathbb{E}\left[w_l x_l\right] + \mathbb{E}\left[b_l\right] = n_l \mathbb{E}\left[w_l\right] \mathbb{E}\left[x_l\right] + \mathbb{E}\left[b_l\right]$$

We can expect $w_l$ and $b_l$ to be zero mean, since many commonly used initializers use either uniform, or normal distribution, with zero mean [28]. This results in:

$$= n_l * 0 * \mathbb{E}\left[x_l\right] + 0 = 0$$

Thus, we can assume that for an individual layer the output is 0, since the final layer of any mlp is 0, we can see that $\mathbb{E}\left[\phi_u(m_{ij})\right] \approx 0$

Next we show that $\mathbb{E}\left[\boldsymbol{u}_i^L\right] \approx \mathbb{E}\left[\boldsymbol{u}_i^0\right]$. Recall that $\boldsymbol{u}_i^L$ is the $L$th layer of $i$th node of the equivariant component of the network. We take the expectation over a sampling of the inputs $u_i$ and $h_i$, treating each as a random variable. Note that $u_i$ and $u_j$ will have identical distributions, since each node will be sampled in the same manner. We start our proof by taking the expected value of the output:

$$\mathbb{E}\left[\boldsymbol{u}_i^{l+1}\right] = \mathbb{E}\left[\boldsymbol{u}_i^l + C\sum_{j \neq i}\left(\boldsymbol{u}_i^l - \boldsymbol{u}_j^l\right)\phi_x\left(\boldsymbol{m}_{ij}\right)\right] = \mathbb{E}\left[\boldsymbol{u}_i^l\right] + C\sum_{j \neq i}\mathbb{E}\left[\left(\boldsymbol{u}_i^l - \boldsymbol{u}_j^l\right)\phi_x\left(\boldsymbol{m}_{ij}\right)\right]$$

$$= \mathbb{E}\left[\boldsymbol{u}_i^l\right] + C\sum_{j \neq i}\mathbb{E}\left[\boldsymbol{u}_i^l\phi_u\left(\boldsymbol{m}_{ij}\right)\right] - \mathbb{E}\left[\boldsymbol{u}_j^l\phi_u\left(\boldsymbol{m}_{ij}\right)\right]$$

$$\approx \mathbb{E}\left[\boldsymbol{u}_i^l\right] + C\sum_{j \neq i}\mathbb{E}\left[\boldsymbol{u}_i^0\right]\mathbb{E}\left[\phi_u\left(\boldsymbol{m}_{ij}\right)\right] - \mathbb{E}\left[\boldsymbol{u}_j^0\right]\mathbb{E}\left[\phi_u\left(\boldsymbol{m}_{ij}\right)\right]$$

$$\approx \mathbb{E}\left[\boldsymbol{u}_i^l\right] + C\sum_{j \neq i}\boldsymbol{u}_i^l 0 - \boldsymbol{u}_j^l 0 = \boldsymbol{u}_i^l$$

$$\mathbb{E}\left[\boldsymbol{u}_i^{l+1}\right] \approx \mathbb{E}\left[\boldsymbol{u}_i^l\right]$$

Thus,

$$\mathbb{E}\left[\boldsymbol{u}_i^L\right] ... \approx \mathbb{E}\left[\boldsymbol{u}_i^1\right] \approx \mathbb{E}\left[\boldsymbol{u}_i^0\right]$$

Where we used the assumption that $\mathbb{E}\left[\boldsymbol{u}_j^l\phi_u\left(\boldsymbol{m}_{ij}\right)\right] \approx \mathbb{E}\left[\boldsymbol{u}_j^l\right]\mathbb{E}\left[\phi_u\left(\boldsymbol{m}_{ij}\right)\right]$

The corollary: $\pi_i(\boldsymbol{a}_i|\boldsymbol{s}) \sim N(\boldsymbol{s}_i^{eq}, \sigma)$ is a simple result from the fact that the equivariant action for an agent is the output of the coordinate embedding of the EGNN: $u_i^L$. Here we assume the standard deviation is a separately trained parameter and not a function of the EGNN. If this is the case, then the policy mean $E\left[\pi(\boldsymbol{a}_i|\boldsymbol{s})\right] = \mathbb{E}\left[\boldsymbol{u}_i^L\right] \approx \mathbb{E}\left[\boldsymbol{u}_i^0\right] = \mathbb{E}\left[\boldsymbol{s}_i^{eq}\right]$ (by definition)

## A.2  Proof E2GN2 is Unbiased

Here we show that E2GN2 leads to unbiased early exploration. We begin with the equation for the E2GN2 layer:

$$\boldsymbol{u}_i^{l+1} = \boldsymbol{u}_i^l \phi_{u2}(\boldsymbol{m}_{ij}) + C \sum_{j \neq i} \left( \boldsymbol{u}_i^l - \boldsymbol{u}_j^l \right) \phi_u \left( \boldsymbol{m}_{ij} \right))$$

Next we take the expected value of the output:

$$E\left[\boldsymbol{u}_i^{l+1}\right] = \mathbb{E}\left[\boldsymbol{u}_i^{l+1}\phi_{u2}(\boldsymbol{m}_{ij})\right] + C \sum_{j \neq i} \mathbb{E}\left[\boldsymbol{u}_i^l \phi_u \left(\boldsymbol{m}_{ij}\right)\right] - \mathbb{E}\left[\boldsymbol{u}_j^l \phi_u \left(\boldsymbol{m}_{ij}\right)\right]$$

$$\approx \mathbb{E}\left[\boldsymbol{u}_i^{l+1}\right] \mathbb{E}\left[\phi_{u2}(\boldsymbol{m}_{ij})\right] + C \sum_{j \neq i} \mathbb{E}\left[\boldsymbol{u}_i^l\right] \mathbb{E}\left[\phi_u \left(\boldsymbol{m}_{ij}\right)\right] - \mathbb{E}\left[\boldsymbol{u}_j^l\right] \mathbb{E}\left[\phi_u \left(\boldsymbol{m}_{ij}\right)\right]$$

$$\approx \mathbb{E}\left[\boldsymbol{u}_i^{l+1}\right] \boldsymbol{0} + C \sum_{j \neq i} \mathbb{E}\left[\boldsymbol{u}_i^l\right] \boldsymbol{0} - \mathbb{E}\left[\boldsymbol{u}_j^l\right] \boldsymbol{0} = \boldsymbol{0}$$

Thus we see that for each layer $\mathbb{E}\left[\boldsymbol{u}_i^{l+1}\right] \approx \boldsymbol{0}$ and therefore $\mathbb{E}\left[\boldsymbol{u}_i^L\right] \approx \boldsymbol{0}$. Similar to before, we used the assumption that Where we used the assumption that $\mathbb{E}\left[\boldsymbol{u}_j^l \phi_u \left(\boldsymbol{m}_{ij}\right)\right] \approx \mathbb{E}\left[\boldsymbol{u}_j^l\right] \mathbb{E}\left[\phi_u \left(\boldsymbol{m}_{ij}\right)\right]$ and $\mathbb{E}\left[\boldsymbol{u}_j^l \phi_{u2}(\boldsymbol{m}_{ij})\right] \approx \mathbb{E}\left[\boldsymbol{u}_j^l\right] \mathbb{E}\left[\phi_{u2}(\boldsymbol{m}_{ij})\right]$

Similar to the first corollary the result: $\pi_i(\boldsymbol{a}_i|\boldsymbol{s}) \sim N(\boldsymbol{0}, \sigma)$ follows mostly from the definition and the above proof. The corollary $\pi_i(\boldsymbol{a}_i|\boldsymbol{s}) \sim N(\boldsymbol{0}, \sigma)$ is a simple result from the fact that the equivariant action for an agent is the output of the coordinate embedding of the EGNN: $u_i^L$. Here we assume the standard deviation is a separately trained parameter and not a function of the EGNN. If this is the case, then the policy mean $E\left[\pi(\boldsymbol{a}_i|\boldsymbol{s})\right] = \mathbb{E}\left[\boldsymbol{u}_i^L\right] \approx \boldsymbol{0}$ (by definition)

## A.3  Equivariance and Invariance of E2GN2

We follow the structure of [2] for how to show equivariance. Note that [2] showed that $\phi_u \left(\boldsymbol{m}_{ij}\right)$ will be invariant to $E(n)$ transformations. This should still hold in our case as we make no modifications to $m_{ij}$, similarly $\phi_{u2} \left(\boldsymbol{m}_{ij}\right)$ will be invariant to translations. To show equivariance to rotations and reflections we show that applying a transformation $T$ to the input, results in a tranformation to the output. That is: $f(\boldsymbol{T}\boldsymbol{u}_i^l) = \boldsymbol{T}\boldsymbol{u}_i^l$ where $f(.)$ is the update equation of E2GN2:

$$\boldsymbol{T}\boldsymbol{u}_i^l \phi_{u2}(\boldsymbol{m}_i) + C \sum_{j \neq i} \left(\boldsymbol{T}\boldsymbol{u}_i^l - \boldsymbol{T}\boldsymbol{u}_j^l\right) \phi_u \left(\boldsymbol{m}_{ij}\right)$$

$$= \boldsymbol{T}\boldsymbol{u}_i^l \phi_{u2}(\boldsymbol{m}_i) + \boldsymbol{T}C \sum_{j \neq i} \left(\boldsymbol{u}_i^l - \boldsymbol{u}_j^l\right) \phi_u \left(\boldsymbol{m}_{ij}\right)$$

$$= \boldsymbol{T}\boldsymbol{u}_i^l \phi_{u2}(\boldsymbol{m}_i) + \boldsymbol{T}C \sum_{j \neq i} \left(\boldsymbol{u}_i^l - \boldsymbol{u}_j^l\right) \phi_u \left(\boldsymbol{m}_{ij}\right)$$

$$= \boldsymbol{T}\left(\boldsymbol{u}_i^l \phi_{u2}(\boldsymbol{m}_i) + C \sum_{j \neq i} \left(\boldsymbol{u}_i^l - \boldsymbol{u}_j^l\right) \phi_u \left(\boldsymbol{m}_{ij}\right)\right) = \boldsymbol{T}\boldsymbol{u}_i^{l+1}$$

Since there are no modifications to E2GN2 update equation for $\boldsymbol{h}_i$, then E2GN2 should retain invariance to transformations from $E(n)$. More precisely (using the fact that the initial $\boldsymbol{m}_{ij}$ is invariant to translations from [2]), if we translate the input $\boldsymbol{u}_i$ by $\boldsymbol{b}$:

$$(\boldsymbol{u}_i^l + \boldsymbol{b})\phi_{u2}(\boldsymbol{m}_i) + C \sum_{j \neq i} \left((\boldsymbol{u}_i^l + \boldsymbol{b}) - (\boldsymbol{u}_j^l + \boldsymbol{b})\right) \phi_u \left(\boldsymbol{m}_{ij}\right)$$

$$= \boldsymbol{b}\phi_{u2}(\boldsymbol{m}_i) + \boldsymbol{u}_i^l \phi_{u2}(\boldsymbol{m}_i) + C \sum_{j \neq i} \left(\boldsymbol{u}_i^l - \boldsymbol{u}_j^l\right) \phi_u \left(\boldsymbol{m}_{ij}\right) = \boldsymbol{b}\phi_{u2}(\boldsymbol{m}_i) + \boldsymbol{u}_i^{l+1} \neq \boldsymbol{b} + \boldsymbol{u}_i^{l+1}$$

This shows that E2GN2 is not translation equivariant.

## B   Additional Training Details

| Hyperparameters | value |
|---|---|
| train batch size | 8000 |
| mini-batch size | 2000 |
| PPO clip | 0.1 |
| learning rate | 25e-5 |
| num SGD iterations | 15 |
| gamma | 0.99 |
| lambda | 0.95 |

Table 3: PPO Hyperparameters for SMACv2

| Hyperparameters | value |
|---|---|
| train batch size | 2000 |
| mini-batch size | 1000 |
| PPO clip | 0.2 |
| learning rate | 30e-5 |
| num SGD iterations | 10 |
| gamma | 0.99 |
| lambda | 0.95 |

Table 4: hyperparameters for MPE

MLP uses two hidden layers of 64 width each for MPE and 128 for SMACv2. All MLPs in the GNNs use 2 layers with a width of 32. For both MPE and GNN structures we use separate networks for the policy and value functions.

**Graph Structure** The graph structure for MPE environments is set as a complete graph. The graph structure for SMACv2 is complete among the agent controlled units (or all of the friendly nodes), and each friendly node alse has an edge to each of the enemy controlled units. This was an attempt to model what we imagine a real world scenario to look like (agents see the enemies, communicate with allies, but do not see the enemies communication signals)

**Graph Inputs** For MPE environments the input invariant feature for each node $h_i^0$ is the id (pursuer, evader, or landmark). For SMACv2, $h_i^0$ is made up of the features: health, shield unit type, and team. For each $u_i^0$ is set as the position of node $i$. For MPE there is also a velocity feature, which we incoporate following the procedure described in [2]

**Graph Outputs for Value function** Similar to the visual in section 4.3, the value function output comes from the invariant component of the agent's node of final layer of the EGNN/E2GN2. In other words the value function is: $h_i^L$ where is $L$ the final layer, and $i$ is the agent in making the decision.

**Graph Outputs for Policy** For MPE environments the actions are forces in a particular direction, thus the action output for agent $i$ is $u_i^L$ where $L$ is the final layer.

Section 4.3 discussed applying EGNN to more sophisticated action spaces. Specifically for SMACv2 the discrete action space has an equivariant component (movements) and an invariant component (shoot). The outputs for the equivariant component of EGNN/E2GN2 are continuous values, mapping these values to logits is not straightforward. Furthermore, they would need to be mapped onto the same distribution of logits being represented by the shoot commands. We solve this problem by having three distributions output by the GNN structure: the continuous/equivariant movement gaussian distribution, the discrete/invariant distribution, and a third distribution that determines whether we should move or shoot. After the RL sampling from each distribution is performed for the actions, then the three components of the action can be converted to the final action for either a mixed discrete-continuous action space or a discrete action space.

**Graph Neural Network details** We used the same GNN as the EGNN paper: Specifically, we use equations 1 and 3, but update equation 1 to be $m_{ij} = \phi(h_i, h_j, x_i, x_j)$. For convenience we rewrite the GNN equations describing a single layer here:

$$\boldsymbol{m}_{ij} = \phi_e\left(\boldsymbol{h}_i^l, \boldsymbol{h}_j^l\right), \quad \boldsymbol{m}_i = \sum_{j \neq i} \boldsymbol{m}_{ij}, \quad \boldsymbol{h}_i^{l+1} = \phi_h\left(\boldsymbol{h}_i^l, \boldsymbol{m}_i\right)$$

Recall that $\phi_h$ and $\phi_e$ are two layer MLPs. Note that the GNN only has one variable $\boldsymbol{h}_i$ per node (compared to the two variables of $\boldsymbol{h}_i$ and $\boldsymbol{u}_i$ of EGNN/E2GN2). The features inputs are similar to an EGNN but with the two invariant/equivariant components concatenated together. The output for GNN will all come from the variable $\boldsymbol{h}$ as there is no need to distinguish between equivariant and invariant outputs (the action may be composed across several nodes, see above and figure 4).

**Hardware** For training hardware, we trained the graph-structured networks using various GPUs. Many were trained on an A100, but that is certainly not necessary, they didn't need that much space. MLPs were trained on CPUs. We typically used 4 rollout workers across 4 CPU threads, so each training run used 5 CPU threads.

**Further Notes/details on SMACv2** SMACv2 ([14]) is a modification of the original SMAC ([29]). Since SMAC had deterministic initialization and static scenarios it turns out policies could memorize the solution. In each scenario, there is a pre-specified number of agents on each team (we use 5 agents for each team). In each scenario, the agents are randomly initialized in a specific formation. For this work, we use the initial position configuration termed "surrounded" (see fig 7). Finally, it is important to note that SMACv2 will randomly sample unit types from 3 different units (for each scenario). We use the same policy network for each agent but have the unit type as an input in the observation space, so each policy must learn to condition its behavior on the unit type.

## C   Supplementary Results

Below we include supplementary results we believe interesting to understanding further E2GN2's performance.

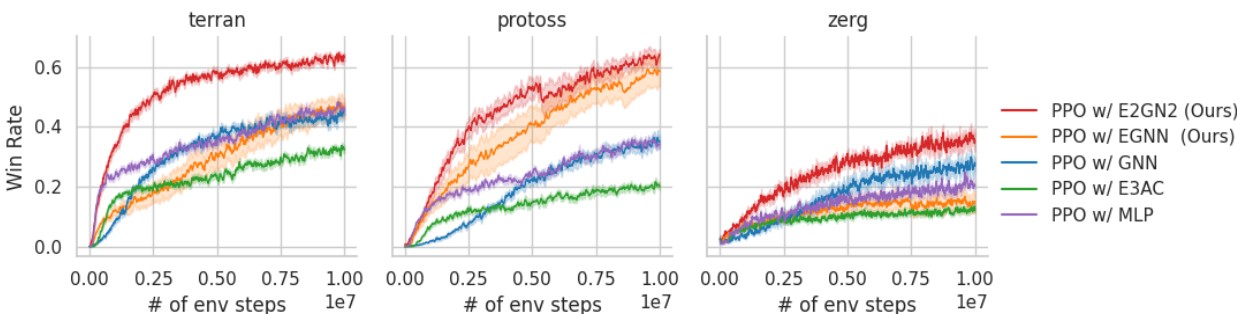

Figure 8: These results are using SMACv2 with the 'surrounded' initialization and 5 agents. Specifically, here we map the actions from the mixed discrete-continuous actions back to the default SMACv2 discrete action space. This is done by simply rounding the continuous actions to the nearest axis. These results are using the standard SMACv2 action input.

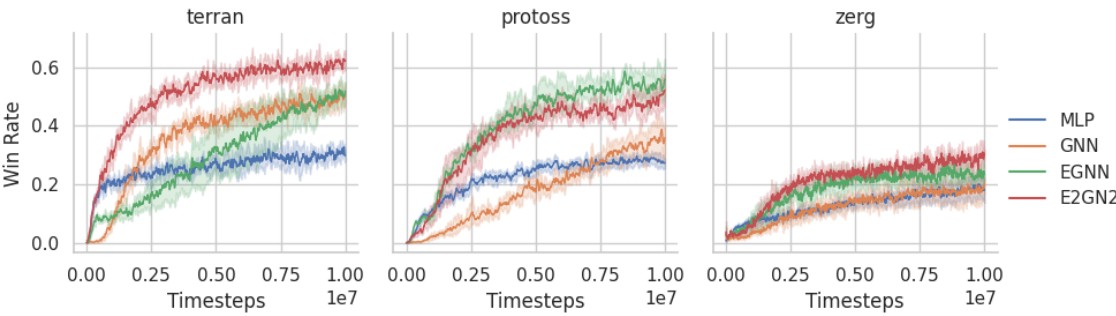

Figure 9: These results are using SMACv2 with the 'surrounded' initialization and 5 agents. Specifically, the observations here are partially observable. Note that E2GN2 still performs well with partial observability.

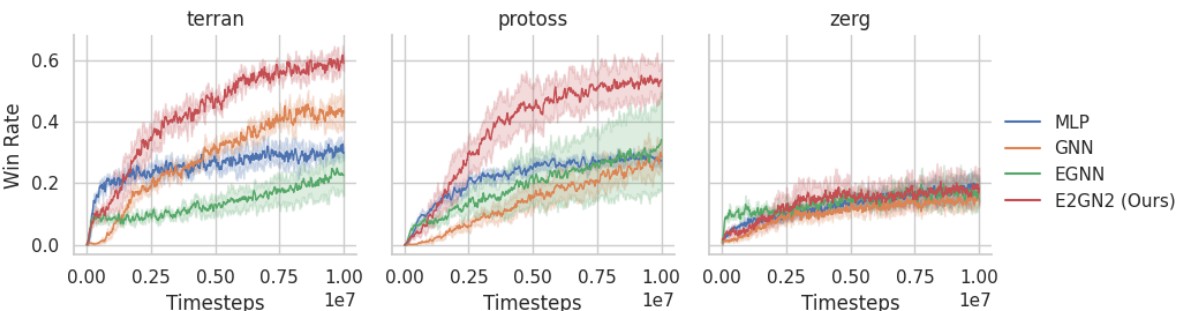

Figure 10: These results are using SMACv2 with the 'surrounded' initialization and 5 agents. Specifically, the observations here are partially observable. In all other experiments we recenter the states at 0 (standard in MARL). Here we did perform the recentering. Note how important this step is for EGNN. Without the centering of observations the EGNN is assuming rotational equivariance around the wrong center. Interestingly, this seems to have little impact on E2GN2's performance.

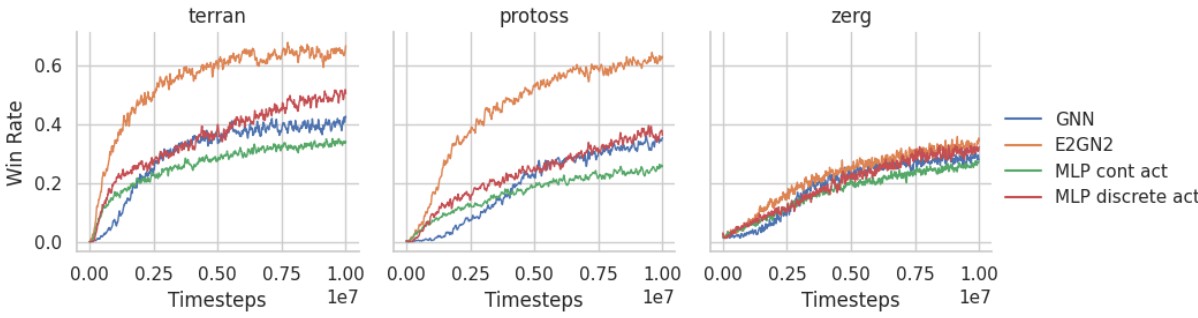

Figure 11: These were trained on SMACv2 using the 'surrounded' initialization with 5 agents. We were curious how an MLP would perform using the mixed continuous/discrete action space vs all discrete. Note that our E2GN2 is still able to outperform both.

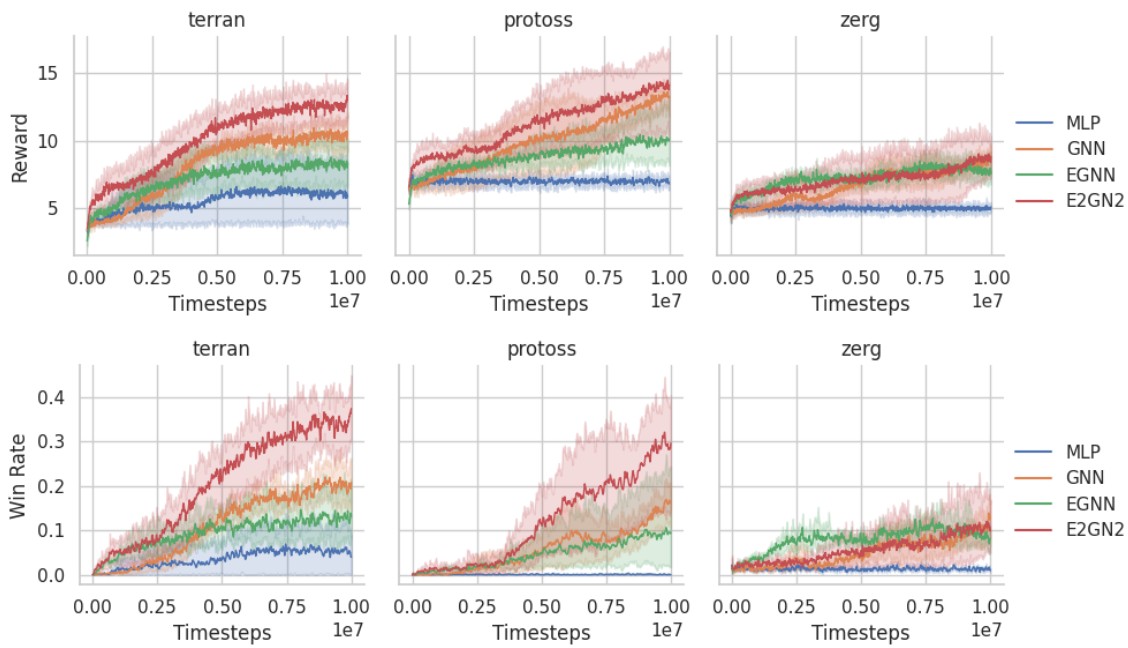

Figure 12: SMACv2 has various initial position configurations. Here we show the results using the 'Surrounded and Reflect' initial configuration.

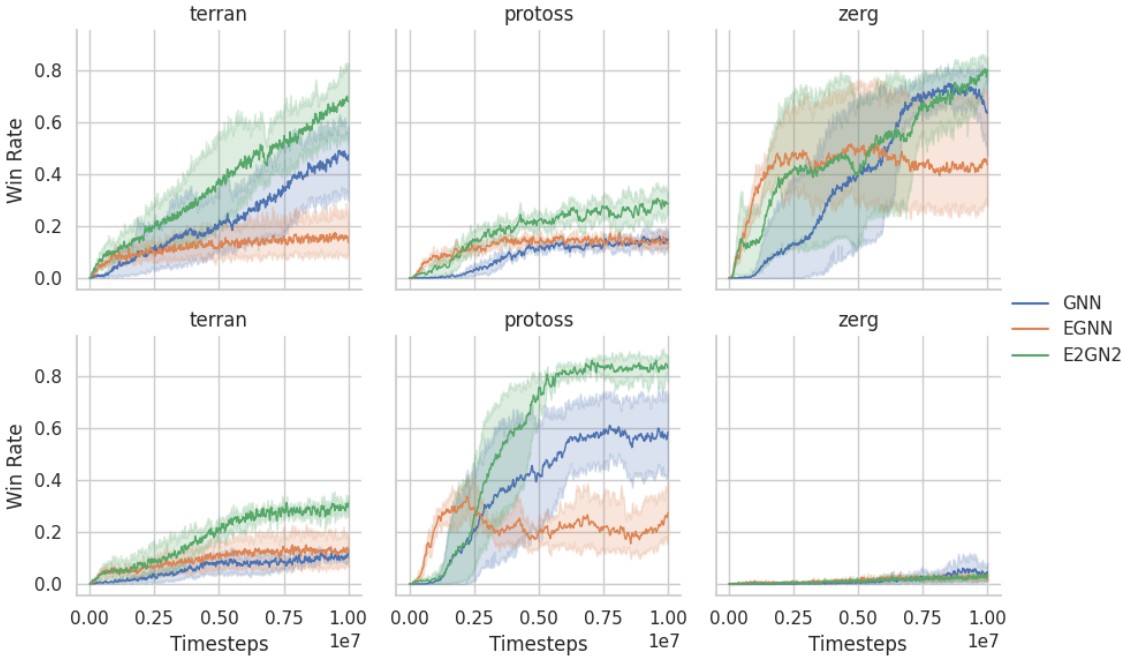

Figure 13: SMACv2, Surrounded and Reflect, 5 agents. Here we dive deeper into the performance when trained on individual unit types (instead of a mix of unit types). Top Row: Only Marine, stalker or Zergling units Bottom Row: Only Marauder, Zealot or hydralisk unit

Table 5: Generalization Win Rate: Training on 5 agents in Surrounded-Left configuration, Testing on 7 agents in all configurations

| Environment | Network | Training Initialization (but with 7 agents) Surrounded Left | Testing Initialization | |
|---|---|---|---|---|
| | | | Surrounded Right | Surrounded All |
| Terran | E2GN2 | 0.362 | 0.354 | 0.432 |
| | GNN | 0.383 | 0.095 | 0.246 |
| Protoss | E2GN2 | 0.431 | 0.418 | 0.424 |
| | GNN | 0.415 | 0.206 | 0.083 |
| Zerg | E2GN2 | 0.207 | 0.156 | 0.145 |
| | GNN | 0.320 | 0.047 | 0.153 |

