# OpenReview forum: "Boosting Sample Efficiency and Generalization in Multi-agent Reinforcement Learning via Equivariance"
_NeurIPS.cc/2024/Conference — NeurIPS 2024 poster_

### Official Review · Reviewer_TKRS · 2024-07-06

**Soundness:** 3
**Presentation:** 3
**Contribution:** 2
**Rating:** 6
**Confidence:** 2

**Summary:**

this paper proposes an exploration-enhanced equivariant graph neural networks. The effective utilization of samples has been achieved in the experimental results. and It  is an improvement on the function approximation for MARL and thus is compatible with most MARL actor-critic methods.

**Strengths:**

The algorithm proposed by the author has been extensively experimented in MPE and SMACv2 environments, and its clear structure is a good paper. And this structure can be applied to other MARL algorithms

**Weaknesses:**

No

**Questions:**

No

---

> ### Author Rebuttal · Authors · 2024-08-07
>
> Thank you for your response and praise in strengths! We added some additional plots in the global response. Let us know if you have any questions regarding those additional results.

---

### Official Review · Reviewer_wBzN · 2024-07-10

**Soundness:** 2
**Presentation:** 3
**Contribution:** 2
**Rating:** 4
**Confidence:** 5

**Summary:**

The paper applies Equivariant Graph Neural Networks (EGNN) to address the issue of low sample efficiency in multi-agent reinforcement learning (MARL). The authors demonstrate that in certain scenarios, EGNN outperforms Multi-Layer Perceptrons (MLP) and Graph Neural Networks (GNN). Key contributions include showcasing EGNN's ability to handle complex action spaces and proposing methods to enhance the exploration capabilities of agents within this framework. The paper presents a limited number of experimental results, attempting to illustrate the superiority of EGNN over some methods in improving exploration efficiency in MARL tasks.

**Strengths:**

1. The authors propose using Equivariant Graph Neural Networks (EGNN) to address the lack of sample efficiency in multi-agent reinforcement learning (MARL), providing insights to promote the application of EGNN in MARL further.
2. They identify the challenge of handling complex action spaces with EGNN and propose a solution to this problem.
3. Highlighting the lack of exploration capability when directly applying EGNN to MARL introduces a new intuition.

**Weaknesses:**

1. The paper lacks baselines that utilize symmetry [1,2,3].
2. There is a lack of entropy-based baselines to compare with for improving exploration capabilities [4].
3. Not all tasks can leverage symmetry, so the paper needs to specify the applicable scope of the proposed method.
4. The innovation is unclear, as extensive work on multi-agent systems is already based on EGNN. Applying EGNN to MARL appears more like trajectory prediction [3]. If the system is controlled solely by displacements in the x and y directions, it essentially remains trajectory prediction, similar to EqMotion [3].
5. The experimental descriptions are insufficient, lacking clarity on how features are designed, such as what constitutes equivariant and invariant features. Additionally, the use of symmetry is closely related to system state transitions, but the dynamics of these transitions are not introduced.
6. The approach to handling partial observability in multi-agent systems is unclear.
7. The reasoning for using zero-mean Gaussian policies is insufficiently explained. Moreover, if the goal is to enhance exploration, it is unclear why entropy-based methods are not used for comparison.
8. Due to the unclear method descriptions and the lack of open-source code, the reproducibility of the paper is poor.
9. Although EGNN can be applied in multi-dimensional spaces, the experiments are all conducted in two dimensions.

[1] Muglich D, Schroeder de Witt C, van der Pol E, et al. Equivariant networks for zero-shot coordination[J]. Advances in Neural Information Processing Systems, 2022, 35: 6410-6423.

[2] Yu X, Shi R, Feng P, et al. Leveraging Partial Symmetry for Multi-Agent Reinforcement Learning[C]//Proceedings of the AAAI Conference on Artificial Intelligence. 2024, 38(16): 17583-17590.

[3] Xu C, Tan R T, Tan Y, et al. Eqmotion: Equivariant multi-agent motion prediction with invariant interaction reasoning[C]//Proceedings of the IEEE/CVF Conference on Computer Vision and Pattern Recognition. 2023: 1410-1420.

[4] Liu I J, Jain U, Yeh R A, et al. Cooperative exploration for multi-agent deep reinforcement learning[C]//International conference on machine learning. PMLR, 2021: 6826-6836.

**Questions:**

1. **Difference from Trajectory Prediction**: How does your application of EGNN in multi-agent reinforcement learning (MARL) fundamentally differ from trajectory prediction in multi-agent systems?
2. **Rationale for Zero-Mean Gaussian Policies**: Why is using zero-mean Gaussian policies considered best practice?
3. **Partial Observability Implementation**: How is partial observability handled in your experiment? Could you provide a detailed explanation of your method?
4. **Enhancing Exploration with Entropy-Based Methods**: Is it possible to incorporate entropy-based methods to enhance exploration in your EGNN framework?

**Limitations:**

The authors need to explore the scope of symmetry utilization to clearly articulate the limitations of their method.

---

> ### Author Rebuttal · Authors · 2024-08-07
>
> Thank you for your response. We have updated our charts, and discuss the novelty further below. We hope you will consider increasing the score given the updates and content.
>
> 1. See the common rebuttal for an additional baseline. [3] is not a MARL algorithm. we attempted to replicate [2], but failed to see any improvement over standard PPO. [1] is only relevant to discrete action spaces and discrete observation spaces. However, we will add citations to [1,3] (we already cite [2]) in the related works section: with the following statement)
>
> 2. Entropy baselines, while potentially helpful are not the standard for MARL (see questions below), and our goal was to compare with standard MARL approaches. Additionally, our contribution was not limited to addressing the exploration bias of EGNN, it was also our contribution to apply EGNN to complex MARL environments leading to improved sample efficiency and generalization.
>
> 3. We believe many tasks will have some form of symmetry, but that symmetry may be inexact, or only partially symmetric. Our approach here cannot currently address those types of problems. However, we are excited to attempt to pursue this in future works.
>
> 4. This seems to be the first work applying EGNN to MARL, and the first work to successfully apply any equivariant neural networks to the common MARL SMACv2 benchmark. See questions section for discussion of trajectory prediction
>
> 5. These are common benchmarks in MARL. We did not design any new features, we used the standard outputs of these environments.
>
> 6. see lines 265-266
>
> 7. See below in questions. for discussion about the gaussian. The goal is not necessarily to only enhance exploration. The goal was to apply Equivariant Neural networks to MARL problems. In the process of using EGNN we noticed that it’s bias was causing a problem in exploration. Thus we had to modify EGNN to improve it’s performance in MARL.
>
> 8. We believe this approach to be fairly reproducible. We knew we may not get permission to release our code. To that end, we specifically used open source libraries for everything we did. The RL code is the open source library RLLIB. We used the code from the EGNN paper for the neural networks. The environments are also open source, as are the features of the environments.
>
> 9. We applied EGNN in 2 dimensions, because our problems (standard benchmarks in MARL literature) were in 2 dimensions. It could certainly be applied to more depending on the problem.
>
>
> Questions
>
> 1. MARL and trajectory prediction are related but fairly different. MARL is attempting to control a set of agents to achieve some objective. This entails learning a value function to approximate the value of each state in achieving that overall objective and a policy to control the agents to achieve that objective. Trajectory prediction is focused on predicting the next step of dynamics. Trajectory prediction is relevant to model-based RL algorithms. but in this case we are focused on model free algorithms
>
> 2. We used the gaussian output for the policy, because that is common practice in the field. Common RL code libraries such as stable baselines https://stable-baselines.readthedocs.io/en/master/, RLLIB all use gaussian parameterized policies. See also: ”...outputting the mean of a Gaussian distribution, with variable standard deviations, following [Sch+15b;
> Dua+16]” from original PPO paper ”The two most common kinds of stochastic policies in deep RL are categorical policies and diagonal Gaussian policies.” https://spinningup.openai.com/en/latest/spinningup/rl intr
> We wanted our approach to be as broadly applicable as possible so wef ocused on common practices.
>
> 3. see lines 265-266
>
> 4. I think that is an interesting idea, and one worth exploring in future efforts.

---

> > ### Comment · Reviewer_wBzN · 2024-08-11
> >
> > After carefully reading and reconsidering the manuscript and the authors' responses to my initial comments, I maintain that the paper requires significant improvements in several key areas:
> >
> > 1.Comparison with Reference [20]: Both this manuscript and reference [20] utilize equivariant neural networks for multi-agent systems. However, reference [20] exhibits greater theoretical depth and technical maturity. Specifically, [20] more thoroughly addresses the symmetry issues in multi-agent systems and utilizes SEGNN, which is more advanced than the EGNN employed in this paper. Consequently, the contributions in terms of innovation and technical depth appear insufficient.
> >
> > 2.Limitations of EGNN: The paper highlights deficiencies in EGNN's exploration performance, a valuable observation. However, the modification proposed in the paper results in the loss of translational equivariance properties. The current treatment and analysis do not adequately resolve or mitigate this issue.
> >
> > 3.Lack of In-depth Analysis on Exploration Performance: Although the paper discusses exploration performance issues, it lacks a thorough comparison with existing entropy-based methods. Such comparisons are essential for a comprehensive evaluation of the method's efficacy presented in the paper. Additionally, the summary of related work is not sufficiently comprehensive, failing to highlight the distinctions and advantages of this research over existing studies.
> >
> > Based on these considerations, I maintain my initial rating. Should the authors address these issues, I will consider raising my score.

---

> > ### Author Response · Authors · 2024-08-12
> >
> > Thank you for your time in reviewing this paper.
> >
> > We believe we addressed these concerns with our rebuttal and responses. In particular, we demonstrated that our approach greatly outperformed [20], with up to 2x the final win rate (on protoss and terran), more than 5x sample efficiency (on protoss and terran), and 4x faster learning time in hours (on MPE). We discussed why translation equivariance is a hindrance to MARL. Finally, we discussed that the MARL literature largely uses gaussian policies and how entropy based MARL exploration is neither state of the art nor common (especially for PPO).
> >
> > Please see our larger official comment to these concerns (posted yesterday) for more details. In your comment you mention you will consider raising your score if we address these concerns, please let us know if there are any other concerns you may have. Thanks!

---

> ### Author Response · Authors · 2024-08-11
>
> Thanks for responding!
>
>
> 1. Note that our paper was written contemporary to [20]. We had much of our results before they published. However, we did add an empirical comparison with [20] and note that our approach E2GN2 (see the pdf in the common rebuttal) greatly outperforms E3AC [20] on all of the SMAC benchmarks. On terran and protoss in particular, we more than double their final win rate (~0.6 vs ~0.3), and have much faster learning: ie E2GN2 learns a win rate of 0.3 in near 1e6 time steps, while E3AC [20] takes 1e7 time steps to learn that win rate. On MPE [20] is much slower, taking 4 hours to train. We believe our improvements are significant (due to using EGNN, E2GN2 and section 4.3) and should be shared with the community.
>
>   Much of the analysis in [20] is only a small extension from prior analysis in Van der pol's work. Van der pol established the theory for adding equivariance to multi-agent MDPs (albiet very simple grid world environments), the work in 20 did not add too much to this in terms of theory. Since van der pol had established this theory we chose not to focus on it as it would not be novel. Instead we focused on more practical applications and improvements.
>
>   Regarding SEGNN, it is not necessarily an improvement over EGNN as it is very slow.  We initially decided to use EGNN specifically because SEGNN and related networks (ie those using spherical harmonics, wigner-D matrices, etc) tend to be slow. Looking at the original SEGNN paper it is 10x slower than EGNN. MARL experiments are already too slow, so we did not want to exacerbate this problem. In our global response we did observe that this results in much slower training times for the results from [20]
>
>   Note that [20] was not able to get competitive results on SMAC. We were able to greatly improve learning on SMAC due to our approach described in section 4.3. Additionally, this solution in section 4.3 is not to only change the movement actions to be continuous, it is to decompose the discrete action outputs such that each node for each enemy is outputting one component of that action. This is crucial for retaining the GNN locality structure and allowing us to scale agents without retraining.
>
>
>
> 2. It is true that E2GN2 loses translation equivariance guarantees. However it retains translation invariance guarantes* see below. For MARL policies translation invariance is useful [see 20] but translation equivariance is harmful.  We certainly observed that in practice translation equivariance was harmful (for one our results demonstrate E2GN2 out performs EGNN). As a simple example. if you have an agent at the position (0,-1) Perhaps the optimal action for agent 1 is to move up: (0,1). Now shift this down by 100, so now this agent's position is at (0,-100). A translation equivariant network will shift the action by -100, leading to the action of (0,-99). This means the agent will always move down, which is not optimal! This is not desirable for many MARL environments. When testing in MPE and SMAC, this may not cause a large problem, as the inputs are often normalized to be near (-1 ,1), but it can still potentially affect performance.
>
> * we can add a full proof to the appendix, on  E2GN2 being translation invariant (just not equivariant), but briefly, we did not modify the equation for computing $h_i$ and are still using the output of $h_i$ for the invariant components.
>
> If we translate the input by $b$, that will translate the output by $\phi_u(m_{ij}) b$ ie   $u_i^l  + \phi_u(m_{ij}) b = u_i^{l-1} + b$ then the output of $h_i^{l+1}$ is translation invariant if  $|| u_i  -u_j  || $ is translation invariant, and $|| u_i + \phi_u(m_{ij}) b  - (u_j^{l} + \phi_u(m_{ij}) b ) || $ = $|| u_i  -u_j  || $ so we retain translation invariance.
>
> 3. For the citation concerns, we have mentioned we will add the citations you mentioned here (as well as others, some reviewers mentioned). We will add the content from these reviews going more in-depth on the differences between our on [19,20,21] (similar to the global rebuttal).  We are not certain we understand the rest of this comment. We were not seeking to change the exploration structure of the MARL learning problem. We were seeking to ameliorate a problem we noted in the EGNN structure (that of bias). We presented an in depth analysis of this bias and compared the EGNN vs E2GN2 across a variety of benchmarks (figure 3,5,6) demonstrating that the exploration bias of EGNN caused worse performance.
>
>
>   There are indeed many methods to attempt to improve a RL algorithm's exploration performance (such as the many papers on curiosity driven exploration). However, our focus was on modifying EGNN to improve common practices in MARL environments/bechmarks (stable baselines, rllib, etc). To that end, we focused on using the common gaussian parameterization of the policy (see our previous comment, Q2 for those citations). EGNN's biased will likely harm any biased exploration.

---

> ### Author Response · Authors · 2024-08-11
>
> 3 continued) Note also that the paper in question [1a] regarding entropy-based exploration does not outperform the benchmarks on dense-rewards, it only outperforms the benchmarks on sparse-reward environments. All of our environments (the standard benchmarks in MARL) have dense rewards, which seems to see little to no improvement from their approach. Furthermore, they do not show how to apply it to actor-critic algorithms such as PPO (what we use here and is one of the best/fastest commonly used algorithms for MARL [2a, rllib])
>
> Entropy-based exploration seems rather niche and has not become a mainstay or mainstream, especially not in MARL. Most of the state of the art papers and libraries use gaussian parameterized policies [2a, rllib, stablebaselines, etc].
>
>
> 1a Cooperative Exploration for Multi-Agent Deep Reinforcement Learning
> 2a The surprising effectiveness of multi-agent PPO

---

### Official Review · Reviewer_2qF5 · 2024-07-11

**Soundness:** 2
**Presentation:** 2
**Contribution:** 2
**Rating:** 4
**Confidence:** 5

**Summary:**

This paper studies an intricate issue, called “early exploration bias”, when applying EGNN, an GNN preserving Euclidean invariance/equivariance, to cooperative MARL that exhibits Euclidean symmetries. The paper reveals that, with randomly initialized weights, the output of the EGNN layers is not centered around zero, which can cause difficulties for cooperative MARL. The paper then proposes a convenient solution to this issue, with a cost of removing translation invariance/equivariance.

**Strengths:**

- The paper is overall easy to follow in terms of its motivation.
- The issue of early exploration bias is well-described using an example (Figure 3), discovering an intricate issue when applying EGNN to MARL.

**Weaknesses:**

- Euclidean equivariance in deep MARL has been explored in several recent works, namely [20] and [Esp: Exploiting symmetry prior for multi-agent reinforcement learning]. This paper’s novelty against these recent works is limited. The most significant contribution seems to be limited to the pathology of EGNN, fixing it to overcome the “early exploration bias” issue. It is unclear whether this pathology and treatment is even relevant in other types of equivariant NNs (e.g., [20] uses the alternative of SEGNN). Moreover, the paper suggests its method can deal with discrete movement actions at the beginning of Section 4.3, which could be a significant novelty from prior work. However, it turns out that the paper bypasses this issue by changing the movement actions in the original SMACv2 from discrete to continuous.

- The method is motivated by the “early exploration bias”. Its cause and the proposed remedy which seems very specific to the proposed usage of EGNN and particularly specific to (MA)RL (e.g., similar issues could arise in ML for chemistry). The paper might benefit from an investigation and discussion of EGNN used in other domains/applications.

- The presentation of the theoretical results (the Theorems and Corollaries) and the proposed methods lacks clarity. Please refer to the Questions on this.

- This paper’s argument on translation equivariance is not that convincing. The proposed method (E2GN2) removes translation invariance/equivariance from EGNN and the paper argues (e.g., Lines 213-218) that translation invariance/equivariance is not a benefit for MARL. Lines 213-218 uses an example to argue against translation equivariance, while previous work (e.g. [20]) suggests translation invariance is indeed benefiting MARL, supported by both theoretical and empirical results. So, the paper could benefit from a revised approach that preserves  translation invariance, not equivariance.

- There are minor issues regarding the quality of writing, e.g.,
    - There is an unwanted line break between lines 110 and 111.
    - Line 122-123: $T_g$ should be $Y \to Y$ and $L_g$ be $X \to X$.
    - Theorem 1: the output vector should be $u_i^{l+1}$.
    - The format of the references is sloppy, e.g. [2, 19, 20, 22]

**Questions:**

1. In Figure 1, why is “All learnable equivariant functions constrained by data” not a subset of  “All learnable equivariant functions”?

2. The original EGNN paper [2] uses it for applications other than RL. Is EGNN’s property presented in Theorem 1 also problematic for the applications there?

3. On the theoretical results:
    - 3a.  In Theorems 1 and 2, what is exactly approximated/hidden in the approximate equalities?
    - 3b.  In Corollary 1.1 and 2.1, can you clarify the use of subscripts $i$ vs $k$?
    - 3c.  In Corollary 1.1 and 2.1, what’s the definition of $s_k^{eq}$?
    - 3d.  In Corollary 1.1 and 2.1, there seems to be a hidden assumption:  $a_i$ has the same dimensionality with $s_k^{eq}$, according to the displayed equation?

4. How exactly is a policy represented by EGNN?
    - 4a. How is the input graph defined/constructed?
    - 4b. What is the dimension of $u^l_i$ for all l?
    - 4c. According to Section 3.3., the dimension of $h^l_i$ is $p$, how to map this $p$-dimensional vector $h_i$ to invariant action components (e.g., logits for the attack targets in SMACv2)? How to ensure such a mapping is equivariant/invariant?

5. The paper does not talk much about value function approximation. How exactly is the value function represented by EGNN?

6. On the experiments:
    - 6a. What specific GNNs are used in the experiments?
    - 6b. The experiments use a decentralized critic (IPPO?). Why not use a centralized critic (MAPPO), which is shown in previous work to perform better?
    - 6c. In Line 287-293, why is “permutation equivariance” relevant? What aspect(s) of the proposed method “allow for” the GNN to scale to larger numbers of agents, in a way that a vanilla GNN cannot?

**Limitations:**

The paper mentions several future work directions, including addressing partial observability, exploring other domains like robotics.

---

> ### Author Rebuttal · Authors · 2024-08-07
>
> Thank you for your thorough, in depth response! Our replies are below:
>
> Weaknesses
> 1.  Note that we added a new comparison with [20] demonstrating ours outperformed the results from [20]. Regarding SEGNN, we initially decided to use EGNN specifically because SEGNN and related networks (ie those using spherical harmonics, wigner-D matrices, etc)  tend to be slow. Looking at the original SEGNN paper it is 10x slower than EGNN. MARL experiments are already too slow, so we did not want to exacerbate this problem.  However, we did add a global response, addressing adding SEGNN from [20] to our comparisons.
>
> Regarding the action space, please see the global rebuttal response. We showed this method is still relevant to SMAC when using discrete actions. Note that E2GN2 and EGNN outperform the baselines in with the discrete action space as well. Our method in 4.3 was crucial for achieving this result.
>
> Note that [20] was not able to get competitive results on SMAC. Additionally, this solution in section 4.3 is not to only change the movement actions to be continuous, it is to decompose the discrete action outputs such that each node for each enemy is outputting one component of that action. This is crucial for retaining the GNN locality structure and allowing us to scale agents without retraining.
>
>
>
> 2. I agree it would be interesting to explore the use case in other domains.. However, our focus in this paper is improving MARL, so perhaps we can explore this in the future.
>
> 3. Okay. We will update our paper accordingly
>
> 4. The results from [20] demonstrate that translation \textit{invariance} is important. Our observation is wrt translation \textit{equivaraince} We certainly observed that in practice translation equivariance was harmful. As a simple example. if you have an agent at the position (0,-1) Perhaps the optimal action for agent 1 is to move up: (0,1). Now shift this down by 100, so now this agent's position is at (0,-100). A translation equivariant network will shift the action by -100, leading to the action of (0,-99). This is not desirable for many MARL environments. When testing in MPE and SMAC, this may not cause a large problem, as the inputs are often normalized to be near (-1,1).
>
> Questions
> 1. Thank you for catching this. The orange circle should say "All learnable functions constrained by data".
>
> 2. This is an interesting question to explore in the future.
>
> 3.  On the theoretical results
>    3b I see how that is confusing. We will update to use just $i$ instead of $k$
>    3c The definition of $s_k^{eq}$ is the component of the action space that is equivariant, we will formalize this definition.
>    3d Yes, that is an assumption that originates with EGNN.
>
>
> 4. How a policy is represented:
>
>     4a) The input graph is described in appendix B (we will change the word choice from "connections" to "edges"). For MPE we use a complete graph. For SMACv2 we use a graph such that it is complete between all friendly nodes. Each friendly node has an edge to each of the enemy nodes. There are no edges between enemy nodes. This was an attempt to model what we imagine a real world type world to look like (you see the enemies, communicate with your allies, but don't see the enemies communications).
>
>    4b) This is partially in appendix B (we reference the width of the MLPs as 32), we can improve the wording by referencing the specific notation, and add details for each environment. For the intermediate GNN layers $u^i_l$ has a dimensionality of 32. For MPE the input $u^0_i$ is the id (pursuer, evader, or landmark). For SMACv2 $u^0_i$ is made up of the features: health, shield unit type, and team.
>
>    4c) This is described in section 4.3 and is key to our approach. For smacv2, the discrete logits for attack targets are the outputs of each node \textit{corresponding to the attack target}. The logit for attacking agent j is the output for $h_j^L$. Since the output of each individual node $h_j^L$ is invariant, these will then remain invariant. These logits are concatenated from each of the nodes to compose the final action. This allows us to scale to larger numbers of agents without retraining.
>
> 5) We did not spend much discussion on the value function approximation, as it has been discussed in prior works [19,21]. The value function is fairly straightforward, we simply used an EGNN/E2GN2 for the value function. The ouput of the own node's invariant component was the value function output ie $u_i^L$ for agent $i$. We will add this to appendix B
>
> 6) On the experiments
>
>      6a) We will add this to the appendix. We used the same GNN as the EGNN paper: Specifically, we use equations 1 and 3, but update equation 1 to be $m_{ij} = \phi(h_i, h_j, x_i,x_j)$.
>
>      6b) A standard GNN layer is permutation equivariant [24]. It also has a local structure that enables adding/removing nodes without needing to retrain the layers. For example, [1a] demonstrates using a GNN to train  N agents, then to control $N+n$ agents without retraining. However, this example did not operate with discrete or mixed action spaces as we do with SMAC.
>
>      Our formulation in 4.3 enables us to retain the permutation equivariance of a GNN.  An alternative to our approach is to add an MLP at the end of the GNN to convert the GNN outputs to the mixed action spaces. This will lose the scalability/locality of the neural network. For example, if you add two more enemies how do you modify the final MLP to expand the action space? In our formulation, whenever an new enemy $i$ is added to the environment, we can simply add the node $N+1$ to the graph. The action space of the GNN will now be supplemented with $u_{N+1}^L$ allowing us to expand the action space without retraining.
>
>
>
> New Citations:
> [1a] Learning Transferable Cooperative Behavior in Multi-Agent Teamhttps://arxiv.org/abs/1906.01202 (GNN)

---

> > ### Comment · Reviewer_2qF5 · 2024-08-09
> >
> > Thanks for your response.
> > It seems that Questions 3a and 6b was not included in your rebuttal, if you would still like to provide a response. Thanks.

---

> ### Author Response · Authors · 2024-08-07
>
> (continuing discussion on point 1 under weaknesses)
>
> To further clarify, the issue with the discrete action space was not necessarily maintaining exact continuous equivariance (we will clarify this in 4.3). The problem is that the  SMACv2 action space has an equivariant component (movements) and an invariant component (shoot). The outputs for the equivariant component of EGNN/E2GN2 are continuous values, mapping these values to logits is not straightforward. Furthermore, they would need to be mapped onto the same distribution of logits being represented by the shoot commands. We solve this problem by having three distributions output by the GNN structure: the continuous/equivariant movement gaussian distribution, the discrete/invariant distribution, and a third distribution that determines whether we should move or shoot (described 290-292). After the RL sampling from each distribution is performed for the actions, then the three components of the action can be converted to the final action for either a mixed discrete-continuous action space or a discrete action space.

---

> > ### Comment · Reviewer_2qF5 · 2024-08-09
> >
> > In your added experiments, did you also use this three-distribution technique for E3AC?
> > Also, is the third distribution (move or shoot) equivariant/invariant?

---

> ### Author Response · Authors · 2024-08-09
>
> Thanks for the reminder. Here are the responses to those points.
>
>
>
>
> 3a. See lines 457 and 469 (I tried copying it here, but the latex kept breaking)
>
> 6b That is a good point to bring up. For clarity, the difference between MAPPO and PPO is that MAPPO uses a centralized critic that takes in the observations and actions of all agents. We believe E2GN2 would improve MAPPO, and we hope to eventually apply this to MAPPO. We use independent PPO because 1) MAPPO performance can vary depending design of the centralized critic [see figure 4 in 1a].  2) For many cases IPPO is comparable to MAPPO [1a]   Since we used parameter sharing [2a], all of our agents were using the same critic. Since we had fully observable environments, the critic had all observations as an input. So using IPPO with parameter sharing and full observability is a fairly close approximation. 3) We wanted to use existing commonly used public libraries to increase reproducibility and utility to the community. We selected RLLIB since it seemed to be a popular benchmark, RLLIB does not have MAPPO built-in, it must be added on your own and implementations can vary (reducing reproducibility)
>
> One more clarification on point 4 of the weaknesses section. E2GN2 is still translation invariant (just not equivariant). We can add a full proof to the appendix, but briefly, we did not modify the equation for computing $h_i$ and are still using the output of $h_i$ for the invariant components.
>
> If we translate the input by $b$, that will translate the output by $\phi_u(m_{ij}) b$ ie   $u_i^l  + \phi_u(m_{ij}) b = u_i^{l-1} + b$ then the output of $h_i^{l+1}$ is translation invariant if  $|| u_i  -u_j  || $ is translation invariant, and $|| u_i + \phi_u(m_{ij}) b  - (u_j^{l} + \phi_u(m_{ij}) b ) || $ = $|| u_i  -u_j  || $ so we retain translation invariance.
>
> 1a https://arxiv.org/abs/2103.01955
> 2a https://arxiv.org/abs/2005.13625

---

> ### Author Response · Authors · 2024-08-09
>
> For E3AC, we used the model and observation processor from the E3AC github repo, so no, E3AC was not using the three-distribution technique we introduced here (since that was part of the novelty of our work). They did not have an observation processor for SMAC in their git repo. We wrote the SMAC observation processor by modifying their MPE observation processor according to the details in their appendix. Note that since E3AC was unable to resolve the difficulty with the SMAC action space (using SEGNN), they opted to use an MLP for the policy and an SEGNN for the value function.
>
> The third distribution is invariant, the logits are output from $h_i^L$ for agent $i$ (ie the invariant part of the output node corresponding to that agent)
>
>  https://github.com/dchen48/E3AC

---

> > ### Comment · Reviewer_2qF5 · 2024-08-11
> >
> > I would raise my rating from 3 to 4, considering the authors have made faithful efforts (especially the additional experiments) to address my comments.
> >
> > However, I am still inclined to recommend a reject, because most of my significant concerns are not yet addressed:
> >
> > The paper made two contributions: (i) correcting EGNN’s idiosyncrasy that causes “early-exploration bias” (sections 4.1 and 4.2)  and (ii) dealing with “complex action spaces” (section 4.3).
> >
> > - For (i), the paper does not justify why we should stick with EGNN.  Switching to a better equivariant architecture should solve the problem. Afterall, the choice of architecture is just a hyperparameter on top of prior work like [20].  The EGNN’s idiosyncrasy is totally independent of MARL, which is why I encouraged the authors to investigate EGNN in other applications/domains.
> >
> > - For (ii), it is a separate contribution from (i), as the “three-distribution” technique can be applied to architectures other than EGNN as well. It seems a valid novelty (prior work like [20] does not address this well), but it seems too incremental to justify an accept.
> >
> > Besides, the paper should benefit from a more clear and rigorous presentation of its theoretical results and method description.

---

> > > ### Author Response · Authors · 2024-08-13
> > >
> > > Regarding the performance of E2GN2 on other applications. We were also curious and tested E2GN2 on the QM9 problem (from EGNN paper). On several QM9 tests, we saw little to no change in performance. We believe this is because the QM9 test predicts invariant quantities. Our modification to EGNN will primarily affect the equivariant component. We did not see an improvement on the n-body trajectory prediction problem either. E2GN2 does not help for trajectory prediction type problems because on those problems, biasing the output by the position is a desirable trait (ie it approximates kinematics, numerical solvers, etc). These were just preliminary tests.
> > >
> > > Furthermore, it is important to consider the translation equivariance property of E2GN2 (again we will add the above discussion on translation equivariance to the appendix). In trajectory prediction translation equivariance is helpful (perhaps why E2GN2 does not improve the n-body task). In MARL translation equivariance is undesirable and harmful. Indeed simply being translation equivariant causes some bias to the network: as an example, for a policy network $\pi(.)$, it is translation equivariant if $\pi(x+b) = \pi(x)+b$, so translation equivariance does introduce a bias to the controls. We believe that is partly why E2GN2 greatly outperforms SEGNN on the Simple Tag problem: an SEGNN policy will have some measure of bias to the output due to the translation equivariance.  Of course, each of these architectures may have more or less bias, depending on each specific formulation. So one of these other architectures may perform better than EGNN, but they will likely have some measure of bias, as they were all designed for a different problem space than MARL/controls.  Although there may be other options, we selected EGNN because it had competitive results on supervised learning benchmarks, and it was much more computationally efficient. Our results and comparisons seem to indicate this was a fruitful decision.

---

> > ### Author Response · Authors · 2024-08-13
> >
> > As the discussion period is coming to a close we would like to thank you again for taking your time in reviewing this paper. We wanted to aggregate our recent comments into a single comment and add a couple of points.
> >
> > To summarize responses regarding the innovations in the paper (see previous comment for more detail)
> >
> > For i: We selected EGNN over other architectures due to computation speed. It takes 40 minutes to train vs SEGNN taking 4 hours for E2GN2 to train, this is why we stick with EGNN and then need to fix the exploration bias. We also note that E2GN2 outperformed both EGNN and SEGNN on MPE Tag.
> >
> > For ii: We believe this novelty is important, even if it is not overly complex. It was critical for using Equivaraint GNNs in a challenging MARL benchmark, and led to enormous improvements in SMAC Terran/protoss. Many AI innovations are simple in concept, but large in impact.
> >
> > SMAC is a much more complex MARL problem due to heterogenous agents with different capabilities, action spaces, etc. It requires complex coordination and cooperation between agents. (see previous comment for more detail).
> > https://www.youtube.com/watch?v=5mUqtGir4e0 https://www.youtube.com/watch?v=VOdiYB3Ut8I (example videos)
> >
> > iii: We believe a third novelty of this paper was applying EGNN to MARL. This did not exist in the literature prior, and EGNN is more feasible than SEGNN (or related formulations) for usage in MARL due to the quicker training time. This may be the key to broader usage of equivariant networks by the MARL community.
> > One quick note (re-novelty) we had much of our results before [20] was published, and did not find it until much of our paper was written.

---

> ### Author Response · Authors · 2024-08-11
>
> Thank you for the response and for upgrading the score! We appreciate you taking the time to carefully consider this work.
>
> To clarify are the significant concerns the two listed in your comment? (i and ii?)
>
> For i: Our greatest rationale for using EGNN (over SEGNN or other architectures) is the speed in computation (see the charts in the main rebuttal pdf) it takes 40 minutes to train vs SEGNN taking 4 hours for E2GN2 to train. This makes a big difference for MARL practitioners. This is why we stick with EGNN and then need to fix the exploration bias. We also note that while SEGNN outperformed EGNN in the original SEGNN paper, here we see that for MPE simple tag EGNN seems to outperform SEGNN.
>
>
> For ii: while this novelty may be fairly straightforward, we have demonstrated it was critical for MARL in scalability and performance on the much more complex SMAC environment (and opening the door for other MARL envs with discrete or mixed action spaces and both equivariant and invariant components). There are many novelties (such as the resnet paper with their skip connections) that were simple in concept, but very important for improving performance, as we see in this paper
>
> iii: We believe a third novelty of this paper was applying EGNN to MARL. This did not exist in the literature prior, and EGNN is more feasible than SEGNN (or related formulations) for usage in MARL due to the quicker training time (in hours). This may be the key to broader usage of equivariant networks by the MARL community.
>
> One quick note (re novelty) we had much of our results before [20] was published, and did not find it until much of our paper was written.

---

> > ### Author Response · Authors · 2024-08-11
> >
> > I wanted to add a little bit more on why SMAC is such a big leap from MPE. MPE has homogenous units with the same capabilities and is primarily focused on cooperative navigation.
> >
> > In SMAC, the units are heterogenous with different capabilities (different attack ranges, total health, and sometimes action spaces). The unit types are randomized at the beginning of the scenario. The actions include more components than simply movement (such as in MPE), agents can move and attack. Some units can heal instead of attack, some units simply explode, others can target multiple enemies.
> > The goals are more complex as well. Instead of simply navigating cooperatively as in MPE, the agents must learn attack formations and strategies. Sometimes it may be optimal to sacrifice health or allies in the purpose of the greater strategic objective.
> >
> > https://www.youtube.com/watch?v=5mUqtGir4e0
> > https://www.youtube.com/watch?v=VOdiYB3Ut8I
> > (example videos)

---

### Official Review · Reviewer_7FjK · 2024-07-12

**Soundness:** 2
**Presentation:** 2
**Contribution:** 3
**Rating:** 6
**Confidence:** 3

**Summary:**

This paper studies the setting of multi-agent reinforcement learning (MARL). The work tries to tackle the challenges of generalization and sample efficiency using inductive biases. In this case, the work proposes to use equivariant graph neural networks to model the policy and value function of a multi-agent actor critic. The work points out a bias in the output of such networks that harms exploration and proposes an architectural fix. It also proposes an architectural modification that will allow the usage of discrete actions in addition to continuous ones. The approach is evaluated on two different benchmarks and in some cases shows improved sample efficiency and final return compared to an MLP and graph neural network baseline.

**Strengths:**

I would like to preface this review by stating that I am not very familiar with the multi-agent RL literature. My estimation of the novelty of this approach has high uncertainty. However, I am very familiar with the standard RL literature.

Problem statement
* Understanding what types of inductive biases can lead to improved sample efficiency in reinforcement learning is in general and interesting and important problem given the high cost of sampling in the real world.

Clarity
* The text is well written and easy to follow.
* The results are presented in a easy to parse manner
* Multiple little example plots are spread throughout the paper to increase readability.

Novelty
* From a brief lit review and given that the work builds on a relatively recent neural network architecture and that other works are concurrently trying to integrate this architecture into MARL agents, I'm willing to believe that fixing the exploration bias and integrating multi-modal action prediction are novel contributions.

Method
* The proposed method uses a well-reasoned fix for a problem in an existing architecture to solve the exploration bias problem. This makes for an interesting investigation.

Related work
* The paper highlights several other works that use equivariance in MARL.

**Weaknesses:**

Clarity
* In the abstract the main claims are 10x improvements in sample efficiency and final reward. This is just unnecessarily exaggerated and possibly misleading. I think the paper would benefit from a clear depiction of what is actually shown.
* The mathematical depiction of the model is unclear and needs to be improved
  * Notation is in places not defined or ill defined. E.g.
    * inputs to equations 1, 2, 3 are unclear to me
    * Some symbols are not defined, e.g. X and Y in section 3.2 The function $\phi_e$ is defined wrt one input in L137 but takes multiple inputs in Eq 1.
  * A viasualization of equations 1, 2, 3 would improve readability. This could have been integrated with Figure 4 to not take up more space.
* Shaded regions in experiments are not explained. I'm assuming they are standard errors.
* In the examples for early exploration, a description of the MDP is required. It was unclear to me what states, actions and rewards are.
* Theorem 2 is missing a pointer to a proof.

Related work
* As mentioned before I am not an expert in the MARL field. The following might not be the most meaningful metric but I figured I'd point it out for completeness. In general, the number of cited papers in the manuscript is low compared to many other works. I'm not going to base my recommendation on this point because a small number of precise citations can be sufficient but I'm pointing this out in case reviewers more familiar with the literature have similar concerns. That is, I don't necessarily expect more citations to be added to raise my score.

Novelty
* The text argues that this work is the first work to successfully demonstrate multi-agent equivariant networks on complex tasks but point to some work whose tasks are not complex. It is unclear to me what the measure of complexity is and when a testbed would be sufficiently complex to make this claim about novelty correct. See Q1.

Experiments and Claims
* Several claims are overstated and need to be adjusted or clarified
  * The work claims that the method outperforms standard GNNs and MLPs by more than 10x sample efficiency. First, this is not validated in the experiments. I do not know what I should look at to get this 10x number. Further, it seems that this is not generally true since in some cases the benefits are marginal (see Figure 6 right) and needs to be rephrased.
  *  In section 5.2, the text states that "equivariant networks have a clear improvement in sample efficiency over the MLP and GNN". This is not fully supported and needs to be concretized. In Figure 6 (right) the statement is incorrect.
* The experiments in Figure 6 are somewhat inconclusive. The final performance of E2GN2 is within variance of EGNN for all three experiments while also being conducted over only 5 seeds. For protoss and zerg, the whole training seems to be within variance, challenging the sample efficiency claim. Increasing statistical significance of these results would make the paper stronger.
* The experimental results in Table 1 and 2 lack a measure of variation. Given that other results seem to be within variance, these measures should be added.
* There is a lack of baselines that compares to prior work. It is, for instance, unclear to me why theoretical guarantees are useful if other approaches might be better. A comparison against the architectures from [19, 21] may have strengthened arguments for being to first to handle complex action-spaces.
* The paper states that the base models were chosen and special tricks from the literature were avoided L255 but that the proposed method "is compatible with most MARL actor-critic methods" L78. First, the paper does not demonstrate that the latter claim is true. Second, I think the paper would have been stronger if it had been shown that the method work *with* these tricks and stronger methods. In general, we have seen (at least in the single agent literature) that combining many advancements is quite beneficial [1 mine, 2 mine, 3 mine]. This would also solve the problem of the baselines being weak.

[1] Combining Improvements in Deep Reinforcement Learning. Matteo Hessel, Joseph Modayil, Hado van Hasselt, Tom Schaul, Georg Ostrovski, Will Dabney, Dan Horgan, Bilal Piot, Mohammad Azar, David Silver. The Thirty-Second AAAI Conference
on Artificial Intelligence, 2017.
[2] Bigger, Better, Faster: Human-level Atari with human-level efficiency. Max Schwarzer, Johan Samir Obando Ceron, Aaron Courville, Marc G Bellemare, Rishabh Agarwal, Pablo Samuel Castro. Proceedings of the 40th International Conference on Machine Learning, PMLR 202:30365-30380, 2023.
[3] Bigger, Regularized, Optimistic: scaling for compute and sample-efficient continuous control. Michal Nauman, Mateusz Ostaszewski, Krzysztof Jankowski, Piotr Miłoś, Marek Cygan. arxiv eprint 2405.16158.

I am happy to slightly raise my score if the language of the claims is adjusted, limitations are addressed and the experiments are made statistically significant. I am happy to raise my score more if additional baselines are included to strengthen claims and the generalizability to stronger approaches is demonstrated.

**Questions:**

Q1: Can you elaborate what makes your environments more complex than the ones used by Van Der Pol et al. [19]?
Q2: Can you elaborate on why previous work such as that by Van Der Pol et al. [19] does not observe the exploration bias?
Q3: How does EGNN in the experiments do action prediction with discrete actions? Does it adapt the suggested solution?

**Limitations:**

The limitations of the work have not been addressed properly. I believe that this would be quite beneficial since the chosen experiments clearly have a structure that benefits from the imposed inductive bias. This does not mean that any claims that are being made can be true **in general**. It is unclear to me whether these methods still function in environments that are not hand-picked for the inductive bias at hand.

---

> ### Author Rebuttal · Authors · 2024-08-07
>
> Thank you for your thorough review. We appreciate the time you took to dive into the details of our work and provide specific advice for improving these results.
>
>
> We tried to take your feedback into account in our updated results. We increased the number of seeds up to 10, and we added a new baseline (see the global rebuttal for those charts) from [20]. We will remove the language regarding "up to 10x improvement in sample efficiency" and replace it with simply "in many cases we see a significant improvement in sample efficiency over GNN and MLP networks". Perhaps that is worded better?
>
> Clarity
> Thank you for this feedback. We will make the recommended changes to the mathematical depiction, improve the definitions for equations 1,2,3 and add the theorem 2 pointer to a proof.
>
> Experiments:
> * On further baselines: We addressed the concerns regarding the baselines [19, 20,21] in the global rebuttal. We added the baseline [20] to our comparisons, and discussed further [19,21]. Let us know if you have any other thoughts on these.
> Before the original submission we did attempt to use [21] as a baseline, but when we tried to implement their approach we did not see any improvement over the standard PPO approach for MPE. (we tried various hyperparameters, and learning rates as well). The experiments in [19] are similar to a grid-world type environment. The input to the traffic control is a 8x8 grid and the cars move one grid per timestep. Their wildlife monitoring environment is a 7x7 grid where the agents move around the grid. Extending their specific approach from a gridworld (discrete number of states) to SMACv2 would likely require further innovations.
>
>
> * On the shaded regions in the experiments: in the submitted paper these regions are the 95% confidence error calculations computed by bootstrapping using python's seaborn. Per your comment on standard errors, the updated plots (see global rebuttal) use standard error instead.
>
> * On the number of seeds, we now have 10 seeds for each run. We note that due to the higher run time of MARL experiments, other MARL papers use around 10 seeds [1a,2a]
>
>
> Q1: On why SMACv2 is more complex: the units are heterogenous with different capabilities (with different attack ranges, total health, and sometimes action spaces). The unit types are randomized at the beginning of the scenario. The actions include more components than simply movement (such as in MPE), agents can move and attack. The goals are more complex as well. Instead of simply navigating cooperatively as in MPE, the agents must learn attack formations and strategies. Sometimes it may be optimal to sacrifice health or allies in the purpose of the greater strategic objective. (see lines 278-286)
>
> Q2: Van der Pol's approach is to find the basis for a specific group (ie rotations). Once the find the basis they make the neural network weights a linear combination of the basis vectors. I don't believe their network would have a bias, but it has the limitation of being applied only to discrete grid-world like environments (see global rebuttal for a further discussion on this).
>
> Q3: In the experiments on SMACv2 (using mixed discrete-continuous actions) EGNN does indeed adapt the suggested solution in 4.3, as does the GNN network. In fact, using the method described in section 4.3 is why GNN was able to increase the number of agents without retraining (Table 2)
>
> Limitations
> We plan to add a further discussion on limitaitons. It is true that the more symmetry in the environment the greater the improvement we would see from EGNN/E2GN2. I will note these environments weren't necessarily hand-picked. MPE and SMAC are the standard/go to environments for MARL. For example, [1a] uses SMAC and MPE and it has 1000 citations. The SMACv2 paper identified problems with the original SMAC environment, which is why we use SMACv2. That said, E2GN2/EGNN could be overly restrictive in environments where symmetries are inexact or unclear. If EGNN is assuming an exact symmetry, but that symmetry is inexact, this could cause a loss in performance as the inductive bias is keeping the network from learning the optimal solution.
>
> Other citations
>
> 1a https://arxiv.org/abs/2103.01955
> 2a https://arxiv.org/abs/1706.05296

---

> ### Author Response · Authors · 2024-08-12
>
> As the discussion period is coming to a close we would like to thank you for your time in the original review. To recap, we have added further discussion on the related works, added more seeds to our experiments, and added a related baseline [20] which our approach continues to outperform. We are eager to hear if our additions have addressed your concerns, and if there is anything else to add to this discussion.
>
> Although we added more seeds to the training charts, we had not yet updated the tables with more seeds (and standard errors). Here we present these:
>
> Table 1 (with 10 training seeds)
>
> | Environment | Network | Surrounded Left  (Training)      | Surrounded Right  (Testing)     | Surrounded All     (Testing)     |
> | ----------- | ------- | ------------- | ------------ | ------------ |
> | terran      | E2GN2   | .57±.01       | .55±.01      | .63±.01      |
> | terran      | GNN     | .43±.02       | .07±.01      | .27±.02      |
> | terran      | MLP     | .33±.02       | .12±.02      | .24 ±.02     |
> | protoss     | E2GN2   | .59±.01       | .56±.02      | .57±.02      |
> | protoss     | GNN     | .44±.02       | .08±.01      | .23±.01      |
> | protoss     | MLP     | .42±.02       | .17±.02      | .27±.02      |
> | zerg        | E2GN2   | .34±.02       | .3±.02       | .31±.02      |
> | zerg        | GNN     | .37±.02       | .06±.01      | .18±.01      |
> | zerg        | MLP     | .24±.02       | .04±.01      | .12±.01      |
>
>
> Table 1 (with 10 training seeds)
> |         |       | 5 Agents (Train) | 4 Agents (Test) | 6 Agents (Test) | 7 Agents (Test) | 8 Agents (Test) |
> | ------- | ----- | ---------------- | --------------- | --------------- | --------------- | --------------- |
> | Terran  | E2GN2 | .69+.02          | .65+.02         | .63+.02         | .62+.02         | .54+.04         |
> |         | GNN   | .48+.02          | .45+.02         | .45+.02         | .40+.02         | .39+.02         |
> | Protoss | E2GN2 | .62+.03          | .61+.02         | .59+.03         | .47+.04         | .37+.03         |
> |         | GNN   | .38+.03          | .24+.02         | .35+.03         | .28+.03         | .2+.02          |
> | Zerg    | E2GN2 | .36+.03          | .32+.03         | .31+.04         | .23+.01         | .18+.03         |
> |         | GNN   | .33+.04          | .29+.03         | .31+.03         | .29+.03         | .27+.03         |
>
>
> Furthermore, one question raised was whether the tips and tricks from [1a] would improve the performance of our approach. First, we clarify that we largely used the majority of their tips and tricks, especially relevant to performance (3 of 5 of their tricks were using a large training batch size, few numbers SGD of iterations, and small clip size. We used all of these).  Due to space in the rebuttal, we couldn't include this explicitly, but if you look at figure 1 from the rebuttal pdf and compare that to figure 5 from the paper, you may notice a difference between the performance of E2GN2 and EGNN. In Figure 1 of the rebuttal, all of the approaches used value normalization from [1a], in the paper that is not necessarily true. In fact, value normalizaiton was critical for E3AC learning at all. Thus we do observe that adding value normalization does indeed improve the results for MPE tag. We did not see an improvement with any other environment (and [1a] didn't see an improvement for other environments either).
>
> Finally, in our blurb on limitations below we discussed how equivariance may be impartial or incomplete in the real world. We also want to add that our approach would need improvements to be applied to environments with angular momentum and mechanics, which are not included in these benchmarks (though to reiterate, these are the standard MARL benchmarks). Note that the total improvement from these approaches will likely depend on the amount of rotational symmetry applicable in the observations. Additionally, this approach is not directly compatible with environments using vision-based observations.
>
>
> We note the reviewer initially wrote they would raise the score if we adjusted the language, addressed limitations, and added more seeds. The reviewer also mentioned raising the score more if we added a new baseline to strengthen the claims and the generalizability to stronger approaches is demonstrated. We hope we addressed your concerns sufficiently, let us know what other concerns you may have.
>
>
> 1a the surprising effectiveness of Multi-agent PPO

---

> > ### Comment · Reviewer_7FjK · 2024-08-12
> >
> > Dear authors,
> >
> > I appreciate the thorough response and am glad you found some of my feedback useful and integrated it.
> >
> > "We will remove the language regarding "up to 10x improvement in sample efficiency" and replace it with simply "in many cases we see a significant improvement in sample efficiency over GNN and MLP networks". Perhaps that is worded better?"
> > Yes, this wording is more appropriate. One could also talk about specific points in time or similar if that feels appropriate. I encourage the authors to make such adjustments where needed so that the claims accurately reflect what is being shown. Thus, my concerns with respect to phrasing of the claims have been partially addressed. I'm saying partially because this really needs to be done for all the claims where required. I'm going to be optimistic and assume that the authors would in fact do this for the final version of the paper even though they have not presented all the rewording here.
> >
> > "Per your comment on standard errors, the updated plots (see global rebuttal) use standard error instead."
> > I was not making a comment about which measure to choose but rather about the fact that you did not explain which one you were using. Bootstrapped confidence is just fine but the text should state that.
> >
> > "On the number of seeds, we now have 10 seeds for each run."
> > These results are more convincing since the gap between methods is more pronounced.
> >
> > For environment complexity, the problem is that I might say: "Previous approaches can not land a rocket on a moon and thus the work did not use sufficiently complex environments." I think being specific about the environment differences in the paper would be beneficial to support the claim of environment complexity.
> > ""Our approach is the first to successfully demonstrate Equivariant GNNs for MARL on standard MARL benchmarks with complex action spaces."
> > The text that the authors point to here does not talk about action or observation spaces.
> > Something like the response to Q1 might be useful to specify what exactly the claim is. I'm going to assume that the authors can include a similar paragraph in the next version of the paper.
> >
> > The new baseline results also put the work into a better perspective with respect to other approaches.
> >
> > Overall, I think the changes improve the presented manuscript significantly and I think at this stage it would be fine to accept this paper. I'm updating my score to 6.

---

> > > ### Author Response · Authors · 2024-08-14
> > >
> > > Thank you again for your review and also for increasing the score. It was very helpful to have specific actionable items to pursue in improving our paper. We believe this review helped improve the paper and strengthen the results.
> > >
> > > Yes, we will update all of the claims to make them more precise and specific. In particular, the abstract, introduction and results section. We will update the comment in the results from  "equivariant networks have a clear improvement in sample efficiency over the MLP and GNN" to "equivariant networks demonstrate improved sample efficiency over the MLP and GNN in the protoss and terran SMACv2 environments".
> > >
> > > After looking at the SMACv2 description in the paper, we agree that the description here is more useful for describing the complexities of this environment. We will add that to the description of SMACv2 in the paper. We will also update references to 'complex action space' or 'complex environments' to be more specific ie: 'mixed discrete-continuous action spaces', and/or 'environments with multi-tiered objectives, that require learning strategies to coordinate heterogenous agents and capabilities'. (Here by multi-tiered objectives we mean that in SMACv2 there are secondary-objectives of killing individual agents and not losing your own agents, vs the overall objectives of winning the scenario)
> > >
> > > Thanks again for your time and effort in contributing to the AI/ML community.

---

### Official Review · Reviewer_N4Ry · 2024-07-14

**Soundness:** 3
**Presentation:** 3
**Contribution:** 3
**Rating:** 6
**Confidence:** 3

**Summary:**

The paper demonstrates for the first time the successful application of Equivariant Graph Neural Networks (EGNNs) to standard MARL benchmarks with complex action spaces.
To address the Early Exploration Bias, the paper provides proof of the biases in standard EGNNs and the unbiased nature of E2GN2. In practice, it is achieved by adding an additional multi-layer perceptron (MLP), which helps to offset the bias from the previous layer's output, leading to actions that are not biased towards the agent's current position.
E2GN2 provides a method to handle complex and heterogeneous action spaces, which are common in MARL environments. It leverages the GNN's graph structure to output different components of the action space from different nodes in an equivariant manner, allowing for a more flexible and scalable approach to handling mixed discrete/continuous action spaces.
The paper shows that E2GN2 outperforms standard GNNs and MLPs by more than 10 times in sample efficiency on common MARL benchmarks like MPE and SMACv2. It also demonstrates greater final reward convergence and up to 5 times gain in generalization over standard GNNs.

**Strengths:**

This paper proposes a novel method with interesting insight and sound theoretical analysis.

**Weaknesses:**

see questions part

**Questions:**

The paper introduces Exploration-enhanced Equivariant Graph Neural Networks (E2GN2), a novel architecture that significantly boosts sample efficiency and generalization in Multi-Agent Reinforcement Learning (MARL) by leveraging equivariance to symmetries and addressing early exploration biases. However, I still have some concerns:
1. For the complex action space issue, the detail of the method is not enough. How do you map the embedding to action space, such as an additional MLP layer? For the value head, does it share the same embedding with the policy head? Section 4.3 and Fig. 4 provide the high-level idea but ignore the details.
2. For the experiment part, this paper seems to use independent ppo as the baseline, which is not widely used in most MARL papers. Although the improvement is significant compared with ppo, it would be better to apply the E2GN2 to other MARL methods.
3. Another issue is the input of the network. To use the EGNN, it seems that the states of all agents are required. However, we assume that only partial observation can be obtained in most MARL settings. For example, we cannot get the observation of opponents in SAMC. Please provide more details of the implementation.

**Limitations:**

see questions

---

> ### Author Rebuttal · Authors · 2024-08-07
>
> Thank you for your review and comments.
>
> 1. There is no MLP on the output of the GNN, EGNN, or E2GN2. Adding an MLP to the output would cause us to lose the guarantee of equivariance as well as the ability to add more agents without retraining.
>
> The output is similar to what is shown in the diagram. It is composed of outputs from various nodes. The outputs all come from the $L$th layer of the network. For agent $i$, the movement component of the action will output from the own node's equivariant component $u_i^L$. For SMAC, the invariant component of the action (ie the logits for determining which target to attack) will be output from the invariant component of that target's node: $h_j^L$ where $j$ is the potential target/enemy. (see lines 232-239 and 290-293).  I hope that helps clarify things.
>
> (we will add this to Appendix B), we use a separate networks for the policy and value function. The value function output is comes from the invariant component of the agent's node of final layer of the EGNN/E2GN2. In other words the value function is: $h_i^L$ Where $L$ is the final layer, $i$ and is the agent in making the decision.
>
> 2. That is a good point to bring up. We believe E2GN2 would improve MAPPO, and we hope to eventually apply this to MAPPO. We use independent PPO because  1) MAPPO can be more difficult for one to one comparisons [1], as it depends on how the centralized critic is shaped. 2) for many cases IPPO is comparable to MAPPO [1] 3) we wanted to use common public libraries for more reliable open source comparisons. we selected RLLIB since it seemed to be a popular benchmark, RLLIB does not have MAPPO built in, it must be added on your own, and implementations can vary.
>
> 3. This is an input variable to SMACv2. A user can change the variable 'partially\_observable' to False to make it fully observable. Note that we did also do a preliminary comparison with partially observable observations (per SMACv2 standard) and saw surprisingly good performance (see appendix)
>
> Citation
> 1) The surprising effectiveness of multi-agent PPO

---

> > ### Author Response · Authors · 2024-08-09
> >
> > One other note regarding point 2
> > Since we used parameter sharing [ ie 2a], all of our agents were using the same critic. Since we had fully observable environments, the critic had all observations as an input. So using IPPO with parameter sharing and full observability is a fairly close approximation.
> >
> > 2a https://arxiv.org/abs/2005.13625

---

### Author Rebuttal · Authors · 2024-08-07

We would like to thank all of their reviewers for their thoughts, comments and advice for improving this research work. We know how busy you all are, and we are grateful you have taken time out of your schedule to provide a thorough and fair review. We are encouraged they found  appreciated our insight into describing and adressing the bias of EGNN in MARL, to improve sample efficiency in MARL.

Based on the below feedback, we added an additional benchmark to our comparisons [20].If accepted we will update the plots with these included here. We will refer to [20] as E3AC (referring to their title).* As before, we ran [20] using RLLIB's PPO to ensure we were using a common benchmark, we trained using the hyperparameters listed in appendix B. Although E3AC is competitive with EGNN on MPE simple spread, it fails to perform as well as E2GN2 in the simple tag. Additionally, although their approach is sample efficient, it is very slow and cumbersome to run. This makes running, tuning and devloping E3AC fairly difficult This is one of the reasons why we chose to use EGNN over other equivarant grpah neural networks: EGNN is much quicker and simpler to run.

Further, as we mentioned in our original paper, E3AC fails to bring the benefits of increased sample efficiency to the more complex problem posed by SMACv2. The results of E3AC are comparable to an MLP. E3AC seems to perform slightly worse on the Zerg environment. We believe this may be due to the fact that some of the zerg agents have more complex action spaces and dynamics (ie some of them explode on contact).

Other benchmarks suggested for comparison include [19, 21]. For [19], their method is specifically formulated and tested on small discrete grid world problems with simple dynamics and discrete up, down,left actions. For example, the trafic control problem has an input of a 7x7 grid. Extending their work to continuous environments with large state spaces, large mixed discrete/continuous action spaces was not straighforward without modifications. For [21], in the past we attempted to replicate their methods , but we failed to see any improvment over PPO on the MPE environments, let alone the more difficult SMAC environments.


Several reviewers were curious about how our results would fare on SMAC with the default discrete action space instead of the mixed continuous discrete space. We present those results here. To map the E2GN2 output to discrete, the continuous movements are mapped back to discrete actions by simply rounding to the nearest discrete action \textit{after} performing exploration (ie sampling from the gaussian distribution). The remaining actions are already discrete. We present those results here as well. We believe this emphasizes the importance of our method described in 4.3, in enabling EGNN structures to mixed discrete-continuous and fully discrete action spaces.

---

### Decision · Program_Chairs · 2024-09-25

**Decision:**

Accept (poster)

**Comment:**

This paper is focused on incorporating equivariance priors into MARL algorithms. This is done by evaluating the already-established EGNN architecture in MARL algorithms. The paper then shows that such an approach has limitations, mainly regarding the exploration strategy it
induces and the discussion around adapting this idea to more complex action spaces. The paper empirically validates its claims on two sets of environments, MPE and SMACv2, showing that EGNN has a slow start (which the paper attributes to exploration issues without necessarily providing data to back that up) and that E2GN2 (the proposed method fixing the exploration issue) has a much better start. The paper also validates its claims around the generalization benefits of using such an inductive bias. Some of the criticisms around the number of independent runs used, and the lack of a specific baseline were addressed to some extent during the rebuttal phase.

The paper has issues, though. One issue that was brought up by pretty much every reviewer is clarity. Many reviewers think the paper is unclear on some of the design choices, and the descriptions provided are insufficient. Although open-sourcing the code is not a requirement for getting a paper accepted at NeurIPS, the lack of clarity added to the impossibility of open-sourcing the code is a major concern. Another major concern is overclaiming. The original version of the paper has very strong statements that are hard to understand where they came from (e.g., 10x improvement). The authors promise to fix the issue, and they provide an example of how they would do so in a specific instance, but it is hard to guarantee that those issues will be fully addressed. There have been many other discussions about relevant related work that should maybe be discussed, or the specific neural network architecture being used, or the scope of the experiments (e.g., partial observability or not, centralized critic or not), which is something that I did not consider so much because I think the paper does a good job defining its scope.

I do feel the pros outweigh the cons here, and I recommend the paper's acceptance. However, I strongly encourage the authors to improve the presentation and adjust their claims.